# AN ADAPTIVE HOMEOSTATIC ALGORITHM FOR THE UNSUPERVISED LEARNING OF VISUAL FEATURES

## ABSTRACT

The formation of structure in the brain, that is, of the connections between cells within neural populations, is by large an unsupervised learning process: the emergence of this architecture is mostly self-organized. In the primary visual cortex of mammals, for example, one may observe during development the formation of cells selective to localized, oriented features. This leads to the development of a rough representation of contours of the retinal image in area V1. We modeled these mechanisms using sparse Hebbian learning algorithms. These algorithms alternate a coding step to encode the information with a learning step to find the proper encoder. A major difficulty faced by these algorithms is to deduce a good representation while knowing immature encoders, and to learn good encoders with a non-optimal representation. To address this problem, we propose to introduce a new regulation process between learning and coding, called homeostasis. Our homeostasis is compatible with a neuro-mimetic architecture and allows for the fast emergence of localized filters sensitive to orientation. The key to this algorithm lies in a simple adaptation mechanism based on non-linear functions that reconciles the antagonistic processes that occur at the coding and learning time scales. We tested this unsupervised algorithm with this homeostasis rule for a range of existing unsupervised learning algorithms coupled with different neural coding algorithms. In addition, we propose a simplification of this optimal homeostasis rule by implementing a simple heuristic on the probability of activation of neurons. Compared to the optimal homeostasis rule, we show that this heuristic allows to implement a more rapid unsupervised learning algorithm while keeping a large part of its effectiveness. These results demonstrate the potential application of such a strategy in machine learning and we illustrate this with one result in a convolutional neural network.

## 1 INTRODUCTION

The neural architecture is a complex dynamic system that operates at different time scales. In particular, one of its properties is to succeed in representing quickly information (the coding step) while optimizing in the long term its encoding (the learning step). In the case of the mammalian primary visual cortex (V1) for instance, this rapid coding operation, of the order of 50 milliseconds in humans, is the key to the results of Hubel & Wiesel (1968), who showed that some cells of V1 have relatively localized receptive fields which are predominantly selective at different orientations. As such, one can consider the rapid coding of the retinal image as a process of transforming the raw visual information into a rough "sketch" that represents the outlines of objects in the image by using elementary edge-like features. This internal representation and the visual information share the same property of being *sparse*: for most natural images, only a relatively small number of features are necessary to describe the input. Thus, the coding step consists in choosing the right encoder that selects as few features (called atoms) as possible among a collection of them (called the *dictionary*). Amazingly, Olshausen & Field (1996) have shown that when enforcing a sparse prior on the encoding step, such edge-like filters are emerging using a simple Hebbian unsupervised learning strategy.

Additionally, recent advances in machine learning, and especially on unsupervised learning have shed new light on the functioning of the underlying biological neural processes. By definition, unsupervised learning aims at learning the best dictionary to represent the input image autonomously, that is, without using other external knowledge such as in supervised or reinforcement learning.

Algorithms that combines such learning as the input to classical, supervised deep-learning show great success in tasks like image denoising (Vincent et al., 2008) or classification (Sulam et al., 2017). A variant consists in forcing the generated representation to be sparsely encoded (Makhzani & Frey, 2013), whether by adding a penalty term to the optimized cost function or by encoding each intermediate representation by a pursuit algorithm (Papyan et al., 2016). Interestingly, (Papyan et al., 2016) proposes a model of Convolutional Sparse Coding (CSC) tightly connected with Convolutional Neural Network (CNN), so much that the forward pass of the CNN is equivalent to a CSC with a thresholding pursuit algorithm. These unsupervised algorithms are equivalent to a gradient descent optimization over an informational-type coding cost (Kingma & Welling, 2013). This cost makes it then possible to *quantitatively* evaluate the joint exploration of new learning or coding strategies. As such, this remark shows us that unsupervised learning consists of two antagonistic mechanisms, a long time scale that corresponds to the learning and exploration of new components and a faster scale that corresponds to coding.

In particular, an aspect often ignored in this type of learning is the set of homeostasis mechanisms that control the average activity of neurons within a population. Indeed, there is an intrinsic complexity in unsupervised dictionary learning algorithms: how to adapt the regularization parameter of each atom to make sure no atoms are wasted because of improper regularization settings? In the original algorithms of sparse unsupervised learning (Olshausen & Field, 1997), homeostasis is implemented as a heuristic that prevents the algorithm from diverging. In most unsupervised learning algorithms it takes the form of a normalization, that is, an equalization of the energy of each atom in the dictionary (Mairal et al., 2014). However, the neural mechanisms of homeostasis are at work in many components of the neural code and are essential to the overall transduction of neural information. For example, the sub-networks of glutamate and GABA-type neurons may regulate the overall activity of neural populations (Marder & Goaillard, 2006). In particular, such mechanisms could be tuned to balance the contribution of the excitatory populations with respect to that in inhibitory populations. As a consequence, this creates a so-called balanced network which may explain many facets of the properties of the primary visual cortex (Hansel & van Vreeswijk, 2012). At the modelling level, these mechanisms are often implemented in the form of normalization rules (Schwartz & Simoncelli, 2001) which are considered as the basis of a normative theory to explain the function of the primary visual cortex (Carandini & Heeger, 2012). However, when extending such model using unsupervised learning, most modelling effort is rather intended to show that the cells' selectivity that emerges have the same characteristics than those observed in neuro-physiology (Ringach, 2002; Rehn & Sommer, 2007; Loxley, 2017). Other algorithms use non-linearities that implicitly implement homeostatic rules in neuro-mimetic algorithms (Brito & Gerstner, 2016). These non-linearities are mainly used in the output of successive layers of deep learning networks that are nowadays widely used for image classification or artificial intelligence. However most of these non-linear normalization rules are based on heuristics mimicking neural mechanisms but are not justified as part of the global problem underlying unsupervised learning. Framing this problem in a probabilistic framework allows to consider in addition to coding and learning the intermediate time scale of homeostasis and allows us also to associate it to an adaptation mechanisms (Rao & Ballard, 1999). Our main argument is that by optimizing unsupervised learning at different time scales, we allow for the implementation of fast algorithms compatible with the performance of biological networks and in comparison with classical (Olshausen & Field, 1997) or Deep Learning approaches.

In this paper, we will first define a simple algorithm for controlling the selection of coefficients in sparse coding algorithms based on a set of non-linear functions similar to a generic neural gain normalization mechanisms. Such functions will be used to implement an homeostasis mechanism based on histogram equalization by progressively adapting these non-linear functions. In particular, this algorithm will extend an already existing algorithm of unsupervised sparse learning (Perrinet, 2010) to a more general setting. In particular, we will show quantitative results of this optimal algorithm by applying it to different pairs of coding and learning algorithms. Second, we will propose a simplification of this homeostasis algorithm based on the activation probability of each neuron and show that it yields similar quantitative results as the full homeostasis algorithm and that it converges more rapidly than classical methods (Olshausen & Field, 1997; Sandin & Martin-del Campo, 2017). In particular, we focused in our architecture to be able to quantitatively cross-validate for every single hyper-parameters and all these scripts are available at `https://github.com/XXX/ZZZ`. Finally, we will conclude by showing an application of such an adaptive algorithm to CNNs and discuss on its development in real-world architectures.

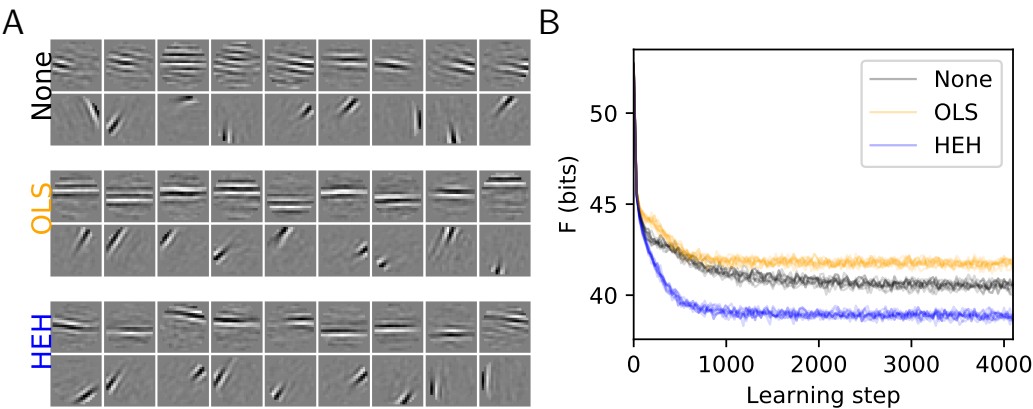

Figure 1: **Role of homeostasis in learning sparse representations**: We show the results of Sparse Hebbian Learning using different homeostasis algorithms at convergence (1024 learning epochs). The compared algorithms are : `None` (using a simple simple normalization of the atoms), `OLS` (the method of (Olshausen & Field, 1997)), `HEH` (using the optimal homeostasis described in this paper). (**A**) For each algorithm, we show 18 atoms from the 441 filters of the same size as the image patches ($M = 22 \times 22 = 484$, circularly masked) and presented in a matrix (separated by a white border). The upper and lower row respectively show the least and most probably selected atoms. This highlights the fact that without proper homeostasis, dictionary learning leads to inhomogeneous representations. (**B**) Evolution of cost $F$ (in bits, see Eq. 4) as a function of the number of iterations and cross-validated over 10 runs. While `OLS` provides a similar convergence than `None`, the `HEH` method provides a better final convergence.

## 2 UNSUPERVISED LEARNING AND THE OPTIMAL REPRESENTATION OF IMAGES

Visual items composing natural images are often sparse, such that knowing a model for the generation of images, the brain may use this property to reconstruct images using only a few of these items. In the context of the representation of natural images[1] $\mathbf{y} = (\mathbf{y}_k)_{k=1}^K \in \mathbb{R}^{K \times M}$ represented in a matrix as a set of $K$ vector samples (herein, we will use a batch size of $K = 256$) as images raveled along $M$ pixels (each $\mathbf{y}_{k,j} \in \mathbb{R}$ are the corresponding luminance values), let us assume the generic Generative Linear Model, such that for any sample $k$ the image was generated as $\mathbf{y}_k = \Phi^T \mathbf{a}_k + \epsilon$, where by definition, the coefficients are denoted by $\mathbf{a}_k = (\mathbf{a}_{k,i})_{i=1}^N \in \mathbb{R}^N$ and the dictionary by $\Phi \in \mathbb{R}^{N \times M}$. Finally, $\epsilon \in \mathbb{R}^M$ is a Gaussian iid noise which is Normal without loss of generality by scaling the norm of the dictionary's rows. Knowing this model, unsupervised learning aims at finding the least surprising causes (the parameters $\hat{\mathbf{a}}_k$ and $\Phi$) for the data $\mathbf{y}_k$. In particular, the cost may be formalized in a probabilistic terms as (Olshausen & Field, 1997):

$$F \approx \langle -\log[p(\mathbf{y}_k|\hat{\mathbf{a}}_k, \Phi)p_\Phi(\hat{\mathbf{a}}_k)]\rangle_{k=1...K} = \langle \frac{1}{2}\|\mathbf{y}_k - \Phi\hat{\mathbf{a}}_k\|_2^2 - \log p_\Phi(\hat{\mathbf{a}}_k)\rangle_{k=1...K} \quad (1)$$

Such hypothesis allows to retrieve the cost that is optimized in most of existing models of unsupervised learning. Explicitly, the representation is optimized by minimizing a cost defined on prior assumptions on representation's sparseness, that is on $\log p_\Phi(\mathbf{a}_k)$. For instance, learning is accomplished in SPARSENET (Olshausen & Field, 1997) by defining a sparse prior probability distribution function for each coefficients in the factorial form $\log p_\Phi(\mathbf{a}_k) \sim -\beta \sum_i \log(1 + \frac{a_i^2}{\sigma^2})$ where $\beta$ corresponds to the steepness of the prior and $\sigma$ to its scaling (see Figure 13.2 from (Olshausen, 2002)). Then, knowing this sparse solution, learning is defined as slowly changing the dictionary using Hebbian learning.

---

[1]We use image patches drawn from large images of outdoor scenes, as provided in the *kodakdb* database which is available in the code's repository.

Indeed, to compute the partial derivate of $F$ with respect to $\Phi$, we have $\forall i$:

$$\frac{\partial}{\partial \Phi_i} F = \langle \frac{1}{2} \frac{\partial}{\partial \Phi_i} [(\mathbf{y}_k - \Phi^T \hat{\mathbf{a}}_k)^T (\mathbf{y}_k - \Phi^T \hat{\mathbf{a}}_k)] \rangle_{k=1...K} = \langle \hat{\mathbf{a}}_k (\mathbf{y}_k - \Phi^T \hat{\mathbf{a}}_k) \rangle_{k=1...K}. \quad (2)$$

This allows to define unsupervised learning as the gradient descent using this equation. Similarly to Eq. 17 in (Olshausen & Field, 1997) or to Eq. 2 in (Smith & Lewicki, 2006), the relation is a linear "Hebbian" rule (Hebb, 1949) since it enhances the weight of neurons proportionally to the activity (coefficients) between pre- and post-synaptic neurons. Note that there is no learning for non-activated coefficients and also that we used a (classical) scheduling of the learning rate and a proper initialization of the weights (see Annex 2.5 & 2.6). The novelty of this formulation compared to other linear Hebbian learning rule such as (Oja, 1982) is to take advantage of the sparse (non-linear) representation, hence the name Sparse Hebbian Learning (SHL). In general, the parameterization of the prior in Eq. 1 has major impacts on results of the sparse coding and thus on the emergence of edge-like receptive fields and requires proper tuning. For instance, a L2-norm penalty term (that is, a Gaussian prior on the coefficients) corresponds to Tikhonov regularization (Tikhonov, 1977) and a L1-norm term (that is, an exponential prior for the coefficients) corresponds to the convex cost which is optimized by least-angle regression (LARS) (Efron et al., 2004) or FISTA (Beck & Teboulle, 2009).

## 2.1 ALGORITHM: SPARSE CODING WITH A CONTROL MECHANISM FOR THE SELECTION OF ATOMS

Concerning the choice of a proper prior distribution, the spiking nature of neural information demonstrates that the transition from an inactive to an active state is far more significant at the coding time scale than smooth changes of the firing rate. This is for instance perfectly illustrated by the binary nature of the neural code in the auditory cortex of rats (DeWeese et al., 2003). Binary codes also emerge as optimal neural codes for rapid signal transmission (Bethge et al., 2003). This is also relevant for neuromorphic systems which transmit discrete, asynchronous events (such as a network packet). With a binary event-based code, the cost is only incremented when a new neuron gets active, regardless to its (analog) value. Stating that an active neuron carries a bounded amount of information of $\lambda$ bits, an upper bound for the representation cost of neural activity on the receiver end is proportional to the count of active neurons, that is, to the $\ell_0$ pseudo-norm $\|\mathbf{a}_k\|_0 = |\{i, \mathbf{a}_{k,i} \neq 0\}|$:

$$F \approx \langle \frac{1}{2} \|\mathbf{y}_k - \Phi \mathbf{a}_k\|_2^2 + \lambda \|\mathbf{a}_k\|_0 \rangle_{k=1...K} \quad (3)$$

This cost is similar with information criteria such as the Akaike Information Criteria (Akaike, 1974) or distortion rate (Mallat, 1998, p. 571). For $\lambda = \log_2 N$, it gives the total information (in bits) to code for the residual (using entropic coding) and the list of spikes' addresses. In general, the high inter-connectivity of neurons (on average approximately 10000 synapses per neurons) justifies such an informational perspective with respect to the analog quantization of information in the point-to-point transfer of information between neurons. However, Eq. 3 defines a harder cost to optimize (in comparison to convex formulations in Equation 1 for instance) since the hard $\ell_0$ pseudo-norm sparseness leads to a non-convex optimization problem which is *NP-complete* with respect to the dimension $M$ of the dictionary (Mallat, 1998, p. 418).

Still, there are many solutions to this optimization problem and here, we will use a generalized version of the Matching Pursuit (MP) algorithm (Mallat, 1998, p. 422). A crucial aspect of this algorithm is the $\arg \max$ function as it produces at each step a competition among $N$ neurons (that is, $\lambda$ bits). For this reason, we will introduce a mechanism to tune this competition. For any signal $\mathbf{y}_k$ drawn from the database, we get the coefficients $\mathbf{a}_k = S(\mathbf{y}_k; \Psi = \{\Phi, z, N_0\})$ thanks to Algorithm 1. The parameter $N_0 > 0$ controls the amount of sparsity that we impose to the coding. The novelty of this generalization of MP lies in the scalar functions $z = \{z_i\}_{i=1...N}$ which control the competition for the best match across atoms. While an identical symmetric function is chosen in the original MP algorithm (that is, $\forall i, z_i(\mathbf{a}_k) = |\mathbf{a}_k|$), we will define these at a first attempt as the rescaled non-linear rectified linear unit (ReLU) with gain $\gamma_i$: $\forall i, z_i(\mathbf{a}_{k,i}) = \gamma_i * \mathbf{a}_{k,i} * \delta(\mathbf{a}_{k,i} > 0)$ where $\delta$ is Kronecker's indicator function.

We found as in (Rehn & Sommer, 2007) that by using an algorithm like Matching Pursuit (that is using the symmetric function or setting $\forall i, \gamma_i = 1$ as in (Mairal et al., 2014) for instance), the Sparse Hebbian Learning algorithm could provide results similar to SPARSENET. An advantage is

---

**Algorithm 1** Generalized Matching Pursuit: $\mathbf{a}_k = S(\mathbf{y}_k; \Psi = \{\Phi, z, N_0\})$

1: set the sparse vector $\mathbf{a}_k$ to zero,
2: initialize $\bar{\mathbf{a}}_{ki} = \langle \mathbf{y}_k, \Phi_i \rangle$ for all $i$
3: **while** $\|\mathbf{a}_k\|_0 < N_0$ **do**:
4:      select the best match: $i^* = \arg\max_i [z_i(\bar{\mathbf{a}}_{ki})]$
5:      update the sparse coefficient: $\mathbf{a}_{k,i^*} = \mathbf{a}_{k,i^*} + \bar{\mathbf{a}}_{k,i^*}$,
6:      update residual coefficients: $\forall i, \bar{\mathbf{a}}_{k,i} \leftarrow \bar{\mathbf{a}}_{k,i} - \mathbf{a}_{k,i^*} \langle \Phi_{i^*}, \Phi_i \rangle$.

---

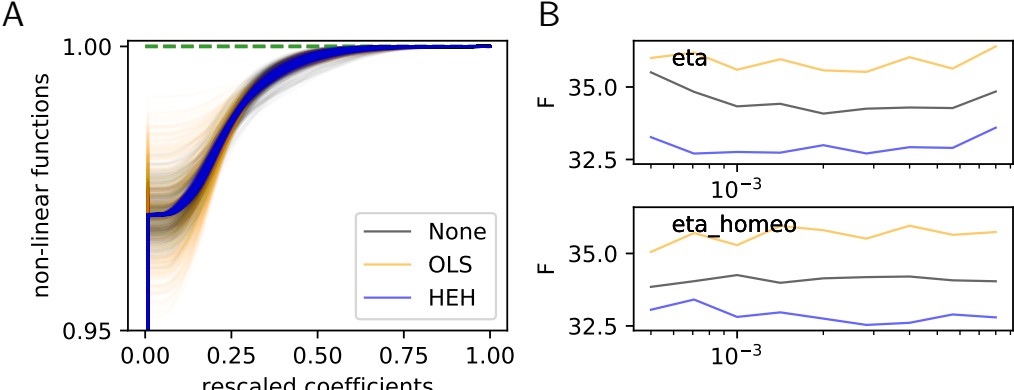

Figure 2: **Histogram Equalization Homeostasis and its role in unsupervised learning**: (A) Non-linear homeostatic functions $z_i, \forall i$ learned using Hebbian learning. These functions were computed for different homeostatic strategies (None, OLS or HEH) but only used in HEH. Note that for our choice of $N_0 = 13$, all cumulative functions start around $1 - N_0/N \approx .970$. At convergence of HEH, the probability of choosing any filter is equiprobable, while the distribution of coefficients is more variable for None and OLS. As a consequence, the distortion between the distributions of sparse coefficients is minimal for HEH, a property which is essential for the optimal representation of signals in distributed networks such as the brain. (B) Effect of learning rate $\eta$ (eta) and homeostatic learning rate $\eta_h$ (eta_homeo) on the final cost as computed for the same learning algorithms but with different homeostatic strategies (None, OLS or HEH). Parameters were explored around a default value, on a logarithmic scale and over 4 octaves. This shows that HEH is robust across a wide range of parameters.

the non-parametric assumption on the prior based on this more generic $\ell_0$ pseudo-norm sparseness. However, we observed that this class of algorithms could lead to solutions corresponding to a local minimum of the full objective function: Some solutions seem as efficient as others for representing the signal but do not represent edge-like features homogeneously (Figure 1-A, None). Moreover, using other sparse coding algorithms which are implemented in the sklearn library, we compared the convergence of the learning with different sparse coding algorithms. In particular, we compared the learning as implemented with matching pursuit to that with orthogonal matching pursuit (OMP) (Pati et al., 1993), LARS or FISTA (see Annex 2.4). For all these sparse coding algorithms, during the early learning step, some cells may learn "faster" than others. In particular, these cells have more peaked distributions of their activity and tend to be selected more often. There is the need for a homeostasis mechanism that will ensure convergence of learning. The goal of this work is to study the specific role of homeostasis in learning sparse representations and to propose a homeostasis mechanism based on the functions $z_i$ which optimizes the learning of an efficient representation.

## 2.2 ALGORITHM: HISTOGRAM EQUALIZATION HOMEOSTASIS

Knowing a dictionary and a sparse coding algorithm, we may transform any data sample $\mathbf{y}_k$ into a set of sparse coefficients using the above algorithm: $\mathbf{a}_k = S(\mathbf{y}_k; \Psi = \{\Phi, z, N_0\})$ (see Algorithm 1). In particular, at any step *during* learning, dictionaries may not have been homogeneously learned and

may exhibit different distributions. However, this would not be taken into account in the original cost (see Eq. 3) as we assumed as in Olshausen & Field (1997) that $p_\Phi(\widehat{\mathbf{a}_k})$ is factorized, that is, that the components of the sparse vector are independent. As a consequence, we may use a deviation to this hypothesis as an additional component to the cost:

$$F \approx \langle \frac{1}{2} \| \mathbf{y}_k - \Phi \mathbf{a}_k \|_2^2 + \lambda \| \mathbf{a}_k \|_0 + \mathtt{MI}(\mathbf{a}_k) \rangle_{k=1\ldots K} \tag{4}$$

Where we used the mutual information $\mathtt{MI}$ as a proxy to measure the dependence between the components of the sparse vector. Indeed, as information is coded in the address of neurons, information transfer as computed through Shannon entropy, is optimized when the activity within the neural population is uniformly balanced, that is when each neuron is *a priori* selected with the same probability. In particular, a necessary (yet not sufficient) condition for minimizing this cost is that the prior probability of selecting coefficients are identical $\forall (i, j), q_\Psi(\mathbf{a}_{k,i}) = q_\Psi(\mathbf{a}_{k,j})$ to ensure the optimality of the choice of the $\ell_0$ pseudo-norm and compare it to the representation in the primary visual cortex. As we have seen, we may use different transformation functions $z$ to influence the choice of coefficients such that we may use these functions to optimize the objective cost defined by Eq. 4.

To achieve this uniformity, we may define an homeostatic gain control mechanism based on histogram equalization, that is, by transforming coefficients in terms of quantiles by setting $\forall i, z_i(\cdot) = P(\cdot > a_i)$. Such a transform is similar to the inverse transform sampling which is used to optimize representation in auto-encoders (Doersch, 2016) and can be considered as a non-parametric extension of the "re-normalization trick" used in variational auto-encoders (Kingma & Welling, 2013). Moreover, it has been found that such an adaptation mechanism is observed in the response of the retina to various contrast distributions (Laughlin, 1981). However, an important point to note is that this joint optimization problem between coding and homeostasis is circular as we can not access the true posterior $p_\Phi(\mathbf{a})$: Indeed, the coefficients depend on non-linear coefficients through $\mathbf{a}_k = S(\mathbf{y}_k; \Psi = \{\Phi, z_i, N_0\})$, while the non-linear functions depend on the (cumulative) distribution of the coefficients. We will make the assumption that such a problem can be solved iteratively by slowly learning the non-linear functions. Starting with an initial set of non-linear functions as in `None`, we will derive an approximation for the sparse coefficients. Then, the function $z_i$ for each coefficient of the sparse vector is calculated using an iterative moving average scheme (parameterized by time constant $1/\eta_h$) to smooth its evolution during learning. At the coding level, this non-linear function is incorporated in the matching step of the matching pursuit algorithm, to modulate the choice of the most probable as that corresponding to the maximal quantile: $i^* = \arg\max_i z_i(a_i)$ (see Algorithm 1). We will coin this variant as Histogram Equalization Homeostasis (HEH). The rest of this Sparse Hebbian Learning algorithm is left unchanged. As we adapt the dictionaries progressively during Sparse Hebbian Learning, we may incorporate this `HEH` homeostasis during learning by choosing an appropriate learning rate $\eta_h$. To recapitulate the different choices we made from the learning to the coding and the homeostasis, the unsupervised learning can be summarized using the following steps (see Algorithm 2).

---

**Algorithm 2** Homeostatic Unsupervised Learning of Kernels: $\Phi = H(\mathbf{y}; \eta, \eta_h, N_0)$

---

1: Initialize the point non-linear gain functions $z_i$ to similar cumulative distribution functions,
2: initialize atoms $\Phi_i$ to random points on the $K$-unit sphere,
3: **for** $T$ epochs **do**:
4:      draw a new batch $\mathbf{y}$ from the database of natural images,
5:      **for** each data point $\mathbf{y}_k$ **do**:
6:          compute the sparse representation vector $\mathbf{a} = S(\mathbf{y}_k; \Psi = \{\Phi, z, N_0\})$ (see Algorithm 1),
7:          modify dictionary: $\forall i, \Phi_i \leftarrow \Phi_i + \eta \cdot \mathbf{a}_i \cdot (\mathbf{y}_k - \Phi \mathbf{a})$,
8:          normalize dictionary: $\forall i, \Phi_i \leftarrow \Phi_i / \| \Phi_i \|$,
9:          update homeostasis functions: $\forall i, z_i(\cdot) \leftarrow (1 - \eta_h) \cdot z_i(\cdot) + \eta_h \cdot \delta(a_i \leq \cdot)$.

---

We compared qualitatively the set $\Phi$ of receptive filters generated with different homeostasis algorithms (see Fig. 1-A). A more quantitative study of the coding is shown by comparing the decrease of the cost as a function of the iteration step (see Fig. 1-B). This demonstrate that forcing the learning activity to be uniformly spread among all receptive fields results in a faster convergence of the representation error as represented by the decrease of the cost $F$.

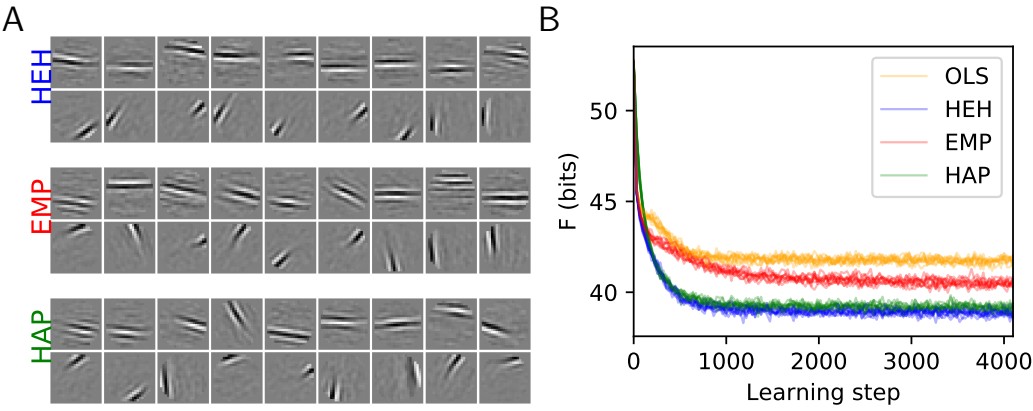

Figure 3: **Homeostasis on Activation Probability (`HAP`) and a quantitative evaluation of homeostatic strategies**: (A) 18 from the 441 dictionaries learned for the two heuristics `EMP` and `HAP` and compared to the optimal homeostasis (see Figure 1-A, `HEH`). Again, the upper and lower row respectively show the least and most probably selected atoms. (B) Comparison of the cost $F$ during learning and cross-validated over 10 runs: The convergence of `OLS` is similar to `EMP`. The simpler `HAP` heuristics gets closer to the more demanding `HEH` homeostatic rule, demonstrating that this heuristic is a good compromise for fast unsupervised learning.

## 2.3 RESULTS: FAST UNSUPERVISED LEARNING USING HOMEOSTASIS

We have shown above that we can find an exact solution to the problem of homeostasis during Sparse Hebbian Learning. However, this solution has several drawbacks. First, it is computationally intensive on a conventional computer as it necessitates to store each $z_i$ function to store the cumulative distribution of each coefficient. More importantly, it seems that biological neurons seem to rather use a simple gain control mechanism. This can be implemented by modifying the gain $\gamma_i$ of the slope of the ReLu function to operate a gradient descent on the cost based on the distribution of each coefficients. Such strategy can be included in the SHL algorithm by replacing line 8 in Algorithm 2. For instance, the strategy of (Olshausen & Field, 1997) assumes a cost on the difference between the observed variance of coefficients $V_i$ as computed over a set of samples compared to a desired value $\sigma_g^2$ (and assuming a multiplicative noise parameterized by $\alpha$) :

$$V_i \leftarrow (1 - \eta_h) \cdot V_i + \eta_h \cdot 1/K \sum_{k=1 \cdots K} a_{i,k}^2 \text{ and } \gamma_i \leftarrow \gamma_i \cdot \left(\frac{V_i}{\sigma_g^2}\right)^\alpha \quad (5)$$

This is similar to the mechanisms of gain normalization proposed by Schwartz & Simoncelli (2001) and which were recently shown to provide efficient coding mechanisms by Simoncelli (2017). However, compared to these methods which manipulate the gain of dictionaries based on the energy of coefficients, we propose to rather use a methodology based on the probability of activation. Indeed, the main distortion that occurs during learning is on higher statistical moments rather than variance, for instance when an atom is winning more at the earlier iterations, its pdf will typically be more kurtotic than a filter that has learned less.

Recently, such an approach was proposed by Sandin & Martin-del Campo (2017). Based on the same observations, the authors propose to optimize the coding during learning by modulating the gain of each dictionary element based on the recent activation history. They base their Equalitarian Matching Pursuit (`EMP`) algorithm on a heuristics which cancels the activation of any filter that was more often activated than a given threshold probability (parameterized by $1 + \alpha_h$). In our setting, we may compute a similar algorithm using an evaluation of probability of activation followed by binary gates:

$$p_i \leftarrow (1 - \eta_h) \cdot p_i + \eta_h \cdot 1/K \sum_{k=1 \cdots K} \delta(a_{i,k} > 0) \text{ and } \gamma_i = \delta(p_i < N_0/N * (1 + \alpha_h)) \quad (6)$$

Interestingly, they reported that such a simple heuristic could improve the learning, deriving a similar result as we have shown in Figure 1 and Figure 2. Again, such strategy can be included in Algorithm 2.

Similarly, we may derive an approximate homeostasis algorithm based on the current activation probability but using a gradient descent approach on gain modulation. Ideally, this corresponds to finding $\gamma_i$ such that we minimize the entropy $-\sum_{i=1\cdots N} p_i \log p_i$. However, the sparse coding function $S(\mathbf{y}_k; \Psi = \{\Phi, z, N_0\})$ is not differentiable. One possible heuristic is then to differentiate the change of modulation gain that would be necessary to achieve an equiprobable probability, that is when $\forall i, p_i = p_0 \overset{\text{def.}}{=} N_0/N$:

$$p_i \leftarrow (1 - \eta_h) \cdot p_i + \eta_h \cdot 1/K \sum_{k=1\cdots K} \delta(a_{i,k} > 0) \text{ and } \gamma_i = \exp(-(p_i - p_0)/\alpha_h) \qquad (7)$$

We will coin this variant of the algorithm Homeostasis on Activation Probability (HAP). Following these derivations, we quantitatively compared OLS, EMP and HAP to HEH (see Figure 3). This shows that while EMP slightly outperforms OLS (which itself is more efficient than None, see Figure 2-B), HAP proves to be closer to the optimal solution given by HEH. In particular, we replicated in HAP the result of Sandin & Martin-del Campo (2017) that while homeostasis was essential in improving unsupervised learning, the coding algorithm (MP versus OMP) mattered relatively little (see Annex 2.4). Also, we verified the dependence of this efficiency with respect to different hyperparameters (as we did in Figure 2-B). These quantitative results show that the HEH algorithm could be replaced by a simpler and more rapid heuristic, HAP, which is based on activation probability. This would generate similar efficiency for the coding of patches from natural images.

## 3    DISCUSSION AND CONCLUSION

One core advantage of sparse representations is the efficient coding of complex signals using compact codes. Inputs are thus represented as combination of few elements drawn from a large dictionary of atoms. As a consequence, a common design for unsupervised learning rules relies on a gradient descent over a cost measuring representation quality with respect to sparseness. This constraint introduces a competition between atoms. In the context of the efficient processing of natural images, we proposed here that such strategies can be optimized by including a proper homeostatic regulation enforcing a fair competition between the elements of the dictionary. We implemented this rule by introducing a non-linear gain normalization similar to what is observed in biological neural networks. We validated this theoretical insight by challenging this adaptive unsupervised learning algorithm with alternate heuristics for homeostasis. Simulations show that at convergence, while the coding accuracy did not vary much, including homeostasis changed qualitatively the learned features. In particular, homeostasis results in a more homogeneous set of orientation selective filters, which is closer to what is found in the visual cortex of mammals (Ringach, 2002; Rehn & Sommer, 2007; Loxley, 2017). To further validate these results, we quantitatively compared the efficiency of the different variants of the algorithms, both at the level of homeostasis (homeostatic learning rate, parameters of the heuristics), but also to the coding (by changing $M$, $N$ or $N_0$) and to the learning (by changing the learning rate, the scheduling or $M$). This demonstrated that overall, this neuro-inspired homeostatic algorithm provided with the best compromise between efficiency and computational cost.

In summary, this biologically-inspired learning rule demonstrates that principles observed in neural computations can help improve real-life machine learning algorithms. Indeed, by developing this fast learning algorithm, we hope for its rapid application in artificial intelligence algorithms. This type of architecture is economical, efficient and fast. It makes it possible to be transferred to most deep learning algorithms. Along with this, we hope that this new type of rapid unsupervised learning algorithm can provide a normative theory for the coding of information in low-level sensory processing, whether it is visual or auditory, for example. Moreover, by its nature, this algorithm can easily be extended to convolutional networks such as those used in deep learning neural networks. This extension is possible by extending the filter dictionary by the hypothesis of invariances to the translation of representations. Our results on different databases show the stable and rapid emergence of characteristic filters on these different bases (see Figure 4 and Annex 3.1). This result shows a probable prospect of extending this representation and for which we hope to obtain classification results superior to the algorithms existing in the state-of-the-art. As such, empirical evaluations of the proposed algorithms should be extended. For instance, it would be very useful to test for image classification results on standard benchmark datasets.

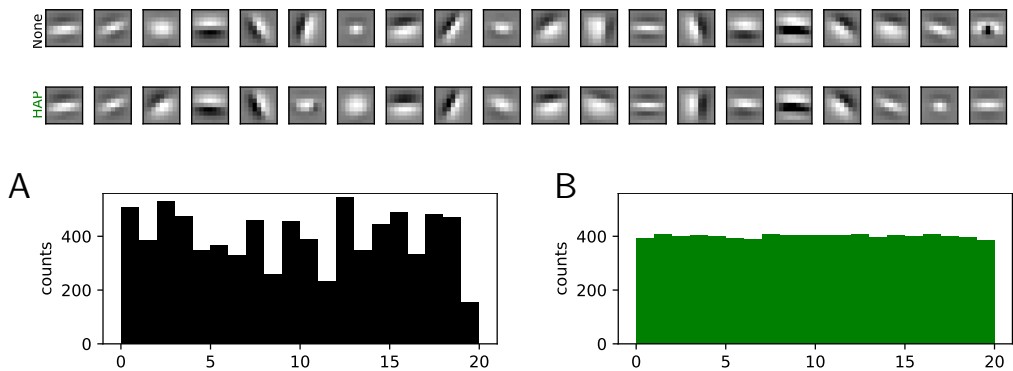

Figure 4: **Extension to Convolutional Neural Networks (CNNs)**: We extend the HAP algorithm to a single layered CNN with 20 kernels and using the ATT face database. We show here the kernels learned without (`None`, top row) and with (`HAP`, bottom row) homeostasis (note that we used the same initial conditions). As for the simpler case, we observe a heterogeneity of activation counts without homeostasis, that is, in the case which simply normalizes the energy of kernels (see (A)). With homeostasis, we observe the convergence of the activation probability for the different kernels (see (B)). This demonstrates that this heuristic extends well to a CNN architecture.

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

This supplementary information presents :

- first, the code to generate the figures from the paper,
- second, some control experiments that were mentionned in the paper,
- finally, some perspectives for future work inspired by the algorithms presented in the paper.

# Figures for "An adaptive algorithm for unsupervised learning"

```
In [1]: %load_ext autoreload
        %autoreload 2
```

```
In [2]: import numpy as np
        np.set_printoptions(precision=2, suppress=True)
        seed = 42
        np.random.seed(seed)
```

```
In [3]: # some overhead for the formatting of figures
        import matplotlib.pyplot as plt

        fontsize = 12
        FORMATS = ['.pdf', '.eps', '.png', '.tiff']
        FORMATS = ['.pdf', '.png']
        dpi_export = 600

        fig_width_pt = 318.670  # Get this from LaTeX using \showthe\column
        width
        fig_width_pt = 450  # Get this from LaTeX using \showthe\columnwidt
        h
        #fig_width_pt = 1024 #221     # Get this from LaTeX using \showthe\
        columnwidth / x264 asks for a multiple of 2
        ppi = 72.27 # (constant) definition of the ppi = points per inch
        inches_per_pt = 1.0/ppi  # Convert pt to inches
        #inches_per_cm = 1./2.54
        fig_width = fig_width_pt*inches_per_pt  # width in inches
        grid_fig_width = 2*fig_width
        phi = (np.sqrt(5) + 1. ) /2
        #legend.fontsize = 8
        #fig_width = 9
        fig_height = fig_width/phi
        figsize = (fig_width, fig_height)

        def adjust_spines(ax, spines):
            for loc, spine in ax.spines.items():
                if loc in spines:
                    spine.set_position(('outward', 10))  # outward by 10 po
        ints
                    spine.set_smart_bounds(True)
                else:
```

```python
                spine.set_color('none')  # don't draw spine

        # turn off ticks where there is no spine
        if 'left' in spines:
            ax.yaxis.set_ticks_position('left')
        else:
            # no yaxis ticks
            ax.yaxis.set_ticks([])

        if 'bottom' in spines:
            ax.xaxis.set_ticks_position('bottom')
        else:
            # no xaxis ticks
            ax.xaxis.set_ticks([])

import matplotlib
pylab_defaults = {
    'font.size': 10,
    'xtick.labelsize':'medium',
    'ytick.labelsize':'medium',
    'text.usetex': False,
#    'font.family' : 'sans-serif',
#    'font.sans-serif' : ['Helvetica'],
    }

#matplotlib.rcParams.update({'font.size': 18, 'font.family': 'STIXG
eneral', 'mathtext.fontset': 'stix'})
matplotlib.rcParams.update(pylab_defaults)
#matplotlib.rcParams.update({'text.usetex': True})

import matplotlib.cm as cm

from IPython.display import Image

DEBUG = True
DEBUG = False
hl, hs = 10*'-', 10*' '
```

In [4]:
```python
tag = 'ICLR'
datapath = '../../SparseHebbianLearning/database'
# different runs
#opts = dict(datapath=datapath, verbose=0)
#opts = dict(cache_dir='cache_dir_cluster', datapath=datapath, verb
ose=0)
#opts = dict(cache_dir='cache_dir_ICLR', datapath=datapath, verbose
=0)
opts = dict(cache_dir='cache_dir_cluster25', eta=0.002, eta_homeo=0
.005, datapath=datapath, verbose=0)
```

In [5]:
```python
from shl_scripts.shl_experiments import SHL
shl = SHL(**opts)
data = shl.get_data(matname=tag)
```

```
In [6]: shl?
```

Type:         SHL
String form: <shl_scripts.shl_experiments.SHL object at 0x10919412
8>
File:
~/science/SparseHebbianLearning/shl_scripts/shl_experiments.py
Docstring:
Base class to define SHL experiments:
    - initialization
    - coding and learning
    - visualization
    - quantitative analysis

```
In [7]: print('number of patches, size of patches = ', data.shape)
        print('average of patches = ', data.mean(), ' +/- ', data.mean(axis
        =1).std())
        SE = np.sqrt(np.mean(data**2, axis=1))
        print('average energy of data = ', SE.mean(), '+/-', SE.std())
```

number of patches, size of patches =  (65520, 324)
average of patches =  5.0641928164665185e-19  +/-  0.0095770518654
37931
average energy of data =  0.29851622590347293 +/- 0.08935954499531
101

```
In [8]: #!ls -l {shl.cache_dir}/{tag}*
        !ls {shl.cache_dir}/{tag}*lock*
        !rm {shl.cache_dir}/{tag}*lock*
        #!rm {shl.cache_dir}/{tag}*
        #!ls -l {shl.cache_dir}/{tag}*
```

ls: cache_dir_cluster25/ICLR*lock*: No such file or directory
rm: cache_dir_cluster25/ICLR*lock*: No such file or directory

# figure 1: Role of homeostasis in learning sparse representations

**TODO : cross-validate with 10 different learnings**

```
In [9]: fname = 'figure_map'
        N_cv = 10
        one_cv = 0 # picking one to display intermediate results
```

### learning

The actual learning is done in a second object (here `dico`) from which we can access another set of properties and functions (see the shl_learn.py
(https://github.com/bicv/SHL_scripts/blob/master/shl_scripts/shl_learn.py) script):

```
In [10]: homeo_methods = ['None', 'OLS', 'HEH']

         list_figures = ['show_dico', 'time_plot_error', 'time_plot_logL', '
         time_plot_MC', 'show_Pcum']
         list_figures = []
         dico = {}
         for i_cv in range(N_cv):
             dico[i_cv] = {}
             for homeo_method in homeo_methods:
                 shl = SHL(homeo_method=homeo_method, seed=seed+i_cv, **opts
         )
                 dico[i_cv][homeo_method] = shl.learn_dico(data=data, list_f
         igures=list_figures, matname=tag + '_' + homeo_method + '_seed=' +
         str(seed+i_cv))

         list_figures = ['show_dico']
         for i_cv in [one_cv]:
             for homeo_method in homeo_methods:
                 print(hl + hs + homeo_method[:3] + hs + hl)
                 shl = SHL(homeo_method=homeo_method, seed=seed+i_cv, **opts
         )
                 shl.learn_dico(data=data, list_figures=list_figures, matnam
         e=tag + '_' + homeo_method + '_seed=' + str(seed+i_cv))

                 print('size of dictionary = (number of filters, size of ima
         gelets) = ', dico[i_cv][homeo_method].dictionary.shape)
                 print('average of filters = ',  dico[i_cv][homeo_method].di
         ctionary.mean(axis=1).mean(),
                       '+/-',  dico[i_cv][homeo_method].dictionary.mean(axis
         =1).std())
                 SE = np.sqrt(np.sum(dico[i_cv][homeo_method].dictionary**2,
         axis=1))
                 print('average energy of filters = ', SE.mean(), '+/-', SE.
         std())
                 plt.show()
```

```
----------          Non          ----------
size of dictionary = (number of filters, size of imagelets) = (44
1, 324)
average of filters =  -1.1980961885168967e-05 +/- 0.00124694840890
4883
average energy of filters =  1.0 +/- 3.920778245506598e-17
```

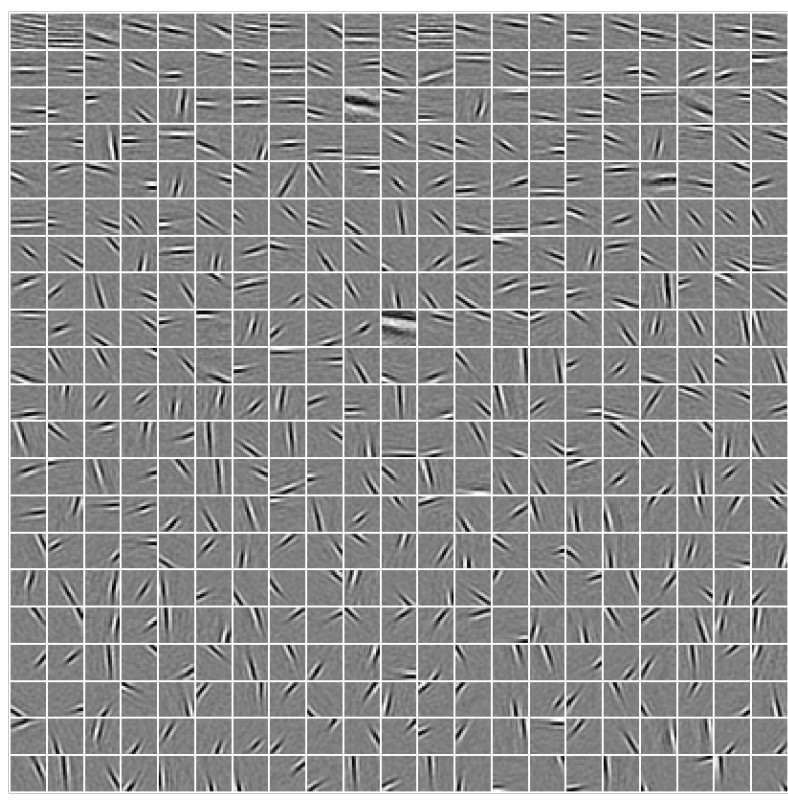

```
----------          OLS          ----------
size of dictionary = (number of filters, size of imagelets) = (44
1, 324)
average of filters =  -4.089243933727358e-06 +/- 0.001241096006797
0878
average energy of filters =  1.0 +/- 3.9562611248144994e-17
```

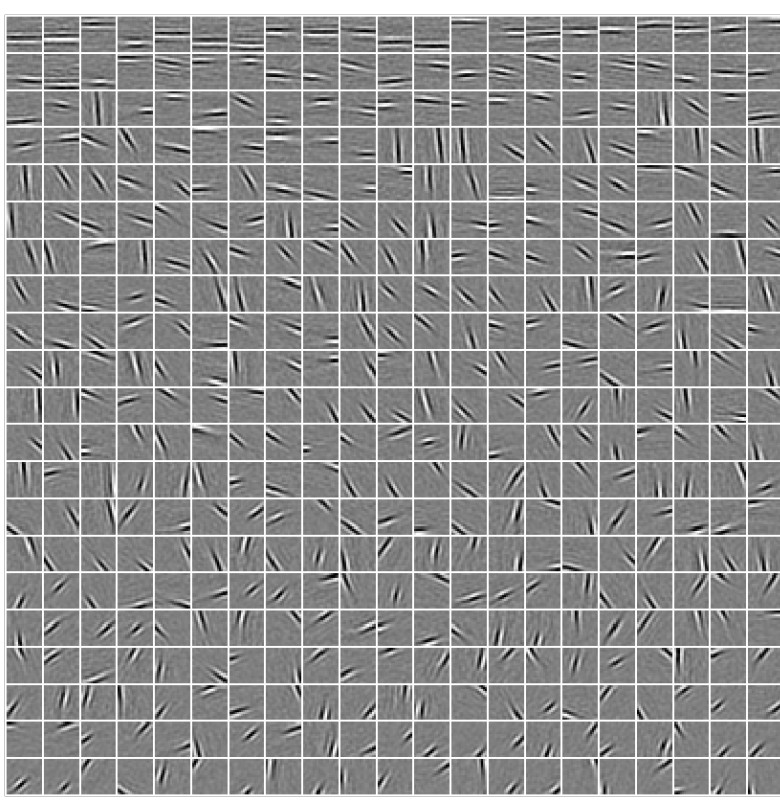

```
----------          HEH          ----------
size of dictionary = (number of filters, size of imagelets) = (44
1, 324)
average of filters =  -6.6112572753952305e-06 +/- 0.00121065448870
92556
average energy of filters =  1.0 +/- 3.700743415417188e-17
```

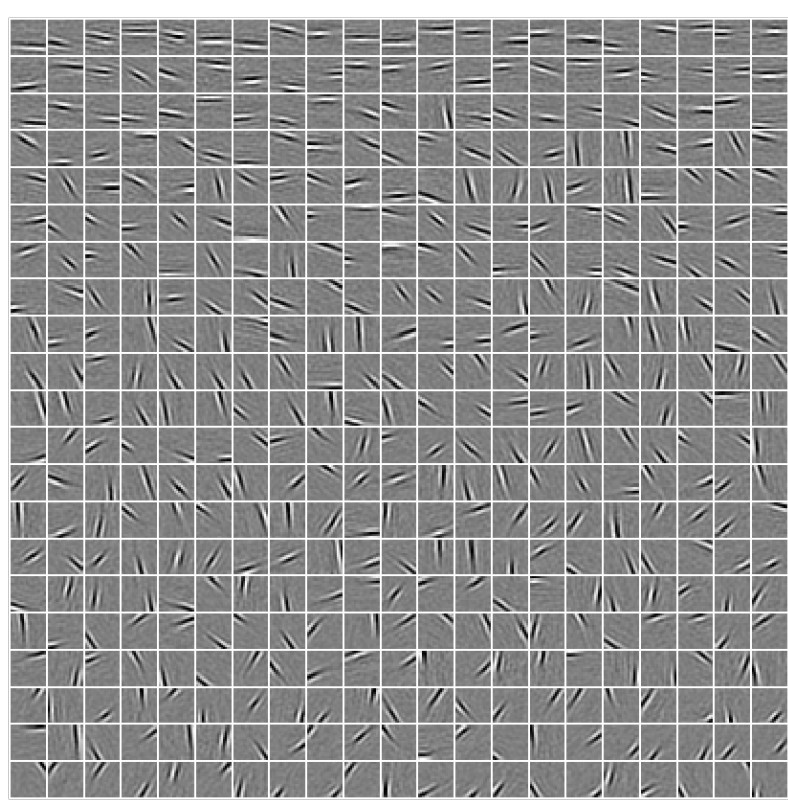

## panel A: plotting some dictionaries

```
In [11]: pname = '/tmp/panel_A' #pname = fname + '_A'
```

```
In [12]: from shl_scripts import show_dico
         if DEBUG: show_dico(shl, dico[one_cvi_cv][homeo_method], data=data,
         dim_graph=(2,5))
```

```
In [13]: dim_graph = (2, 9)
         colors = ['black', 'orange', 'blue']
         homeo_methods
```

```
Out[13]: ['None', 'OLS', 'HEH']
```

```
In [14]: subplotpars = dict( left=0.042, right=1., bottom=0., top=1., wspace
         =0.05, hspace=0.05,)
         fig, axs = plt.subplots(3, 1, figsize=(fig_width/2, fig_width/(1+ph
         i)), gridspec_kw=subplotpars)

         for ax, color, homeo_method in zip(axs.ravel(), colors, homeo_metho
         ds):
             ax.axis(c=color, lw=2, axisbg='w')
             ax.set_facecolor('w')
             fig, ax = show_dico(shl, dico[one_cv][homeo_method], data=data,
         dim_graph=dim_graph, fig=fig, ax=ax)
             # ax.set_ylabel(homeo_method)
             ax.text(-8, 7*dim_graph[0], homeo_method, fontsize=12, color=co
         lor, rotation=90)#, backgroundcolor='white'

         for ext in FORMATS: fig.savefig(pname + ext, dpi=dpi_export)
```

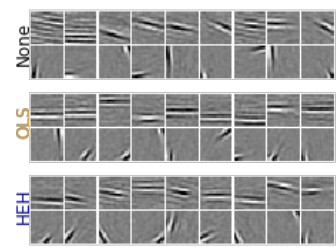

```
In [15]: ### TODO put the p_min an p_max value in the filter map
```

```
In [16]: if DEBUG: Image(pname +'.png')
```

```
In [17]: if DEBUG: help(fig.subplots_adjust)
```

```
In [18]: if DEBUG: help(plt.subplots)
```

```
In [19]: if DEBUG: help(matplotlib.gridspec.GridSpec)
```

### panel B: quantitative comparison

```
In [20]: pname = '/tmp/panel_B' #fname + '_B'
```

```
In [21]: from shl_scripts import time_plot
         variable = 'F'
         alpha_0, alpha = .3, .15
         subplotpars = dict(left=0.2, right=.95, bottom=0.2, top=.95)#, wspa
         ce=0.05, hspace=0.05,)
         fig, ax = plt.subplots(1, 1, figsize=(fig_width/2, fig_width/(1+phi
         )), gridspec_kw=subplotpars)
         for i_cv in range(N_cv):
             for color, homeo_method in zip(colors, homeo_methods):
                 ax.axis(c='b', lw=2, axisbg='w')
                 ax.set_facecolor('w')
                 if i_cv==0:
                     fig, ax = time_plot(shl, dico[i_cv][homeo_method], vari
         able=variable, unit='bits', color=color, label=homeo_method, alpha=
         alpha_0, fig=fig, ax=ax)
                 else:
                     fig, ax = time_plot(shl, dico[i_cv][homeo_method], vari
         able=variable, unit='bits', color=color, alpha=alpha, fig=fig, ax=a
         x)
                 # ax.set_ylabel(homeo_method)
                 #ax.text(-8, 7*dim_graph[0], homeo_method, fontsize=12, col
         or='k', rotation=90)#, backgroundcolor='white'
         ax.legend(loc='best')
         for ext in FORMATS: fig.savefig(pname + ext, dpi=dpi_export)
         if DEBUG: Image(pname +'.png')
```

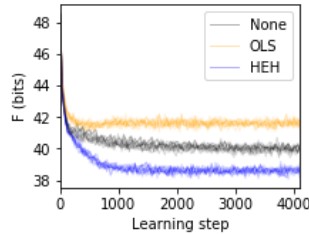

## Montage of the subplots

```
In [22]: import tikzmagic
```

```
In [23]: %load_ext tikzmagic
```

```
In [24]: #DEBUG = True
         if DEBUG: help(tikzmagic)
```

%tikz \draw (0,0) rectangle (1,1);%%tikz --save {fname}.pdf \draw[white, fill=white] (0.\linewidth,0) rectangle (1.\linewidth, .382\linewidth) ;

In [25]:
```
%%tikz -f pdf --save {fname}.pdf
\draw[white, fill=white] (0.\linewidth,0) rectangle (1.\linewidth,
.382\linewidth) ;
\draw [anchor=north west] (.0\linewidth, .382\linewidth) node {\inc
ludegraphics[width=.5\linewidth]{/tmp/panel_A}};
\draw [anchor=north west] (.5\linewidth, .382\linewidth) node {\inc
ludegraphics[width=.5\linewidth]{/tmp/panel_B}};
\begin{scope}[font=\bf\sffamily\large]
\draw [anchor=west,fill=white] (.0\linewidth, .382\linewidth) node
[above right=-3mm] {$\mathsf{A}$};
\draw [anchor=west,fill=white] (.53\linewidth, .382\linewidth) node
[above right=-3mm] {$\mathsf{B}$};
\end{scope}
```

In [26]:
```
!convert  -density {dpi_export} {fname}.pdf {fname}.jpg
!convert  -density {dpi_export} {fname}.pdf {fname}.png
#!convert  -density {dpi_export} -resize 5400  -units pixelsperinch
-flatten  -compress lzw  -depth 8 {fname}.pdf {fname}.tiff
Image(fname +'.png')
```

Out[26]:

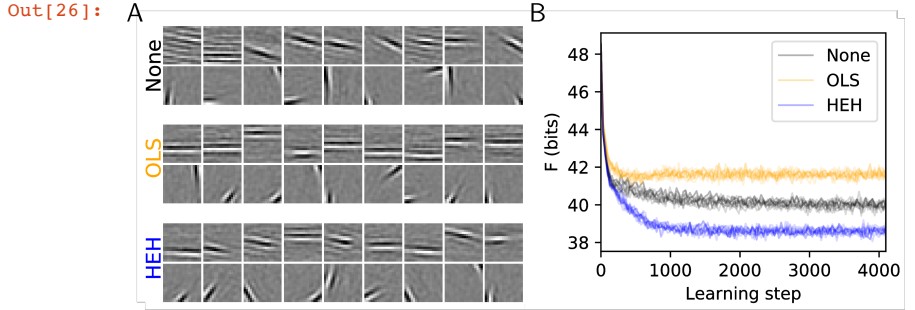

!echo "width=" ; convert {fname}.tiff -format "%[fx:w]" info: !echo ", \nheight=" ; convert {fname}.tiff -format "%[fx:h]" info: !echo ", \nunit=" ; convert {fname}.tiff -format "%U" info:!identify {fname}.tiff

## figure 2: Histogram Equalization Homeostasis

In [27]:
```
fname = 'figure_HEH'
```

First collecting data:

```
list_figures = ['show_Pcum']

dico = {}
for homeo_method in homeo_methods:
    print(hl + hs + homeo_method + hs + hl)
    shl = SHL(homeo_method=homeo_method, **opts)
    #dico[homeo_method] = shl.learn_dico(data=data, list_figures=li
st_figures, matname=tag + '_' + homeo_method + '_' + str(one_cv))
    dico[homeo_method] = shl.learn_dico(data=data, list_figures=lis
t_figures, matname=tag + '_' + homeo_method + '_seed=' + str(seed+o
ne_cv))
    plt.show()
```

---------- None ----------

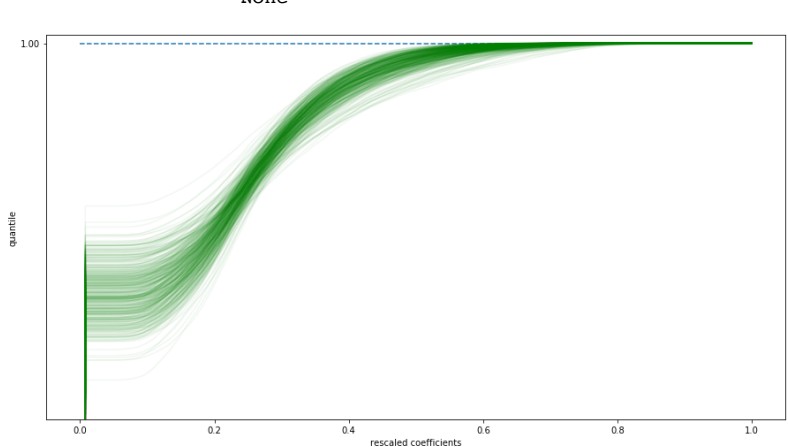

---------- OLS ----------

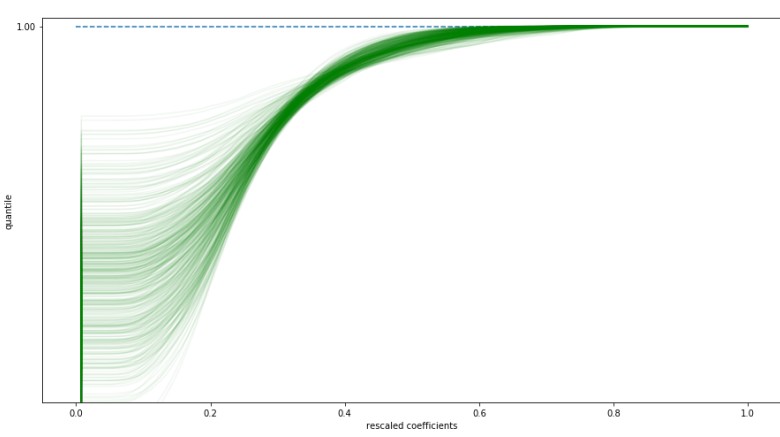

---------- HEH ----------

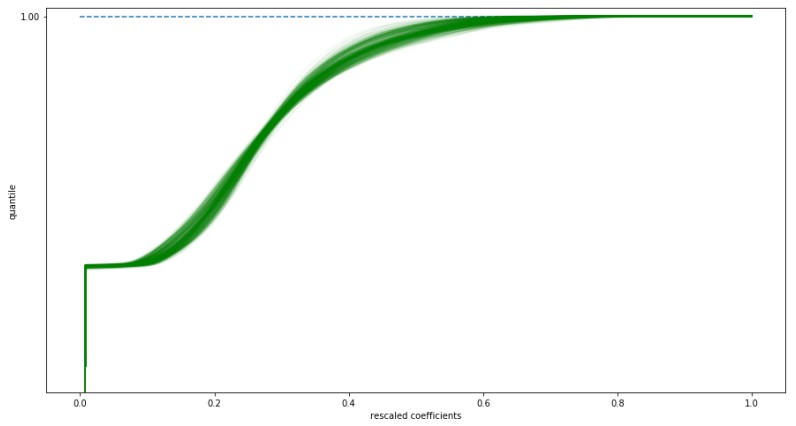

```
In [29]: dico[homeo_method].P_cum.shape
```

Out[29]: (441, 128)

**panel A: different P_cum**

```
In [30]: pname = '/tmp/panel_A' #pname = fname + '_A'

         from shl_scripts import plot_P_cum
         variable = 'F'
         subplotpars = dict(left=0.2, right=.95, bottom=0.2, top=.95)#, wspa
         ce=0.05, hspace=0.05,)
         fig, ax = plt.subplots(1, 1, figsize=(fig_width/2, fig_width/(1+phi
         )), gridspec_kw=subplotpars)
         for color, homeo_method in zip(colors, homeo_methods):
             ax.axis(c='b', lw=2, axisbg='w')
             ax.set_facecolor('w')
             fig, ax = plot_P_cum(dico[homeo_method].P_cum, ymin=0.95, ymax=
         1.001,
                                  title=None, suptitle=None, ylabel='non-lin
         ear functions',
                                  verbose=False, n_yticks=21, alpha=.02, c=c
         olor, fig=fig, ax=ax)
             ax.plot([0], [0], lw=1, color=color, label=homeo_method, alpha=
         .6)
             # ax.set_ylabel(homeo_method)
             #ax.text(-8, 7*dim_graph[0], homeo_method, fontsize=12, color='
         k', rotation=90)#, backgroundcolor='white'
         ax.legend(loc='lower right')
         for ext in FORMATS: fig.savefig(pname + ext, dpi=dpi_export)
         if DEBUG: Image(pname +'.png')
```

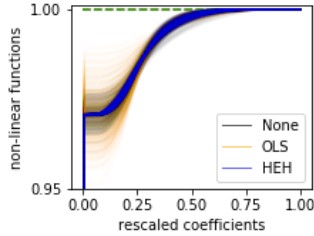

```
In [31]: if DEBUG: help(fig.legend)
```

### panel B: comparing the effects of parameters

```
In [35]: opts
```

```
Out[35]: {'cache_dir': 'cache_dir_cluster25',
          'eta': 0.002,
          'eta_homeo': 0.005,
          'datapath': '../../SparseHebbianLearning/database',
          'verbose': 0}
```

```
In [ ]: pname = '/tmp/panel_B' #fname + '_B'

        from shl_scripts.shl_experiments import SHL_set
```

```python
homeo_methods = ['None', 'EMP', 'HAP', 'HEH', 'OLS']

homeo_methods = ['None', 'OLS', 'HEH']

variables = ['eta', 'alpha_homeo', 'eta_homeo', 'l0_sparseness', 'n
_dictionary']
variables = ['eta', 'alpha_homeo', 'eta_homeo', 'l0_sparseness']
variables = ['alpha_homeo', 'eta_homeo']
variables = ['eta', 'alpha_homeo', 'eta_homeo']
variables = ['eta', 'eta_homeo']

list_figures = []

bases = [10, 10, 2, 2]
bases = [4] * 4

for homeo_method, base in zip(homeo_methods, bases):
    opts_ = opts.copy()
    opts_.update(homeo_method=homeo_method)
    experiments = SHL_set(opts_, tag=tag + '_' + homeo_method)#, ba
se=base)
    experiments.run(variables=variables, n_jobs=1, verbose=0)

import matplotlib.pyplot as plt
subplotpars = dict(left=0.2, right=.95, bottom=0.2, top=.95, wspace
=0.5, hspace=0.35,)

x, y = .05, -.3

if len(variables)==4:
    fig, axs = plt.subplots(2, 2, figsize=(fig_width/2, fig_width/(
1+phi)), gridspec_kw=subplotpars, sharey=True)
    for i_ax, variable in enumerate(variables):
        for color, homeo_method in zip(colors, homeo_methods):
            opts_ = opts.copy()
            opts_.update(homeo_method=homeo_method)
            experiments = SHL_set(opts_, tag=tag + '_' + homeo_meth
od)#, base=base)
            ax = axs[i_ax%2][i_ax//2]
            fig, ax = experiments.scan(variable=variable, list_figu
res=[], display='final', fig=fig, ax=ax, color=color, display_varia
ble='F', verbose=0) #, label=homeo_metho
            ax.set_xlabel('') #variable
            ax.text(x, y,  variable, transform=axs[i_ax].transAxes)
            #axs[i_ax].get_xaxis().set_major_formatter(matplotlib.t
icker.ScalarFormatter())

else:
    fig, axs = plt.subplots(len(variables), 1, figsize=(fig_width/2
, fig_width/(1+phi)), gridspec_kw=subplotpars, sharey=True)

    for i_ax, variable in enumerate(variables):
        for color, homeo_method in zip(colors, homeo_methods):
            opts_ = opts.copy()
            opts_.update(homeo_method=homeo_method)
```

```
             experiments = SHL_set(opts_, tag=tag + '_' + homeo_meth
    od)#, base=base)
             fig, axs[i_ax] = experiments.scan(variable=variable, li
    st_figures=[], display='final', fig=fig, ax=axs[i_ax], color=color,
    display_variable='F', verbose=0) #, label=homeo_metho
             axs[i_ax].set_xlabel('') #variable
             axs[i_ax].text(x, y,  variable, transform=axs[i_ax].tra
    nsAxes)
             #axs[i_ax].get_xaxis().set_major_formatter(matplotlib.t
    icker.ScalarFormatter())

    #fig.legend(loc='lower right')
    for ext in FORMATS: fig.savefig(pname + ext, dpi=dpi_export)
    if DEBUG: Image(pname +'.png')
```

### Montage of the subplots

```
In [ ]: %%tikz -f pdf --save {fname}.pdf
\draw[white, fill=white] (0.\linewidth,0) rectangle (1.\linewidth,
.382\linewidth) ;
\draw [anchor=north west] (.0\linewidth, .382\linewidth) node {\inc
ludegraphics[width=.5\linewidth]{/tmp/panel_A.pdf}};
\draw [anchor=north west] (.5\linewidth, .382\linewidth) node {\inc
ludegraphics[width=.5\linewidth]{/tmp/panel_B.pdf}};
\begin{scope}[font=\bf\sffamily\large]
\draw [anchor=west,fill=white] (.0\linewidth, .382\linewidth) node
[above right=-3mm] {$\mathsf{A}$};
\draw [anchor=west,fill=white] (.53\linewidth, .382\linewidth) node
[above right=-3mm] {$\mathsf{B}$};
\end{scope}
```

```
In [ ]: !convert  -density {dpi_export} {fname}.pdf {fname}.jpg
!convert  -density {dpi_export} {fname}.pdf {fname}.png
#!convert  -density {dpi_export} -resize 5400  -units pixelsperinch
-flatten -compress lzw  -depth 8 {fname}.pdf {fname}.tiff
Image(fname +'.png')
```

!echo "width=" ; convert {fname}.tiff -format "%[fx:w]" info: !echo ", \nheight=" ; convert {fname}.tiff -format
"%[fx:h]" info: !echo ", \nunit=" ; convert {fname}.tiff -format "%U" info:!identify {fname}.tiff

## figure 3:

### learning

```
In [36]: fname = 'figure_HAP'
```

```
In [37]: colors = ['orange', 'red', 'green', 'blue']
         homeo_methods = ['OLS', 'HEH', 'EMP', 'HAP']
         list_figures = []
         dico = {}
         for i_cv in range(N_cv):
             dico[i_cv] = {}
             for homeo_method in homeo_methods:
                 shl = SHL(homeo_method=homeo_method, seed=seed+i_cv, **opts
         )
                 dico[i_cv][homeo_method] = shl.learn_dico(data=data, list_f
         igures=list_figures, matname=tag + '_' + homeo_method + '_seed=' +
         str(seed+i_cv))

         list_figures = ['show_dico'] if DEBUG else []
         for i_cv in [one_cv]:
             for homeo_method in homeo_methods:
                 print(hl + hs + homeo_method + hs + hl)
                 shl = SHL(homeo_method=homeo_method, seed=seed+i_cv, **opts
         )
                 shl.learn_dico(data=data, list_figures=list_figures, matnam
         e=tag + '_' + homeo_method + '_seed=' + str(seed+i_cv))
                 plt.show()
                 print('size of dictionary = (number of filters, size of ima
         gelets) = ', dico[i_cv][homeo_method].dictionary.shape)
                 print('average of filters = ',  dico[i_cv][homeo_method].di
         ctionary.mean(axis=1).mean(),
                       '+/-',  dico[i_cv][homeo_method].dictionary.mean(axis
         =1).std())
                 SE = np.sqrt(np.sum(dico[i_cv][homeo_method].dictionary**2,
         axis=1))
                 print('average energy of filters = ', SE.mean(), '+/-', SE.
         std())
```

```
----------          OLS          ----------
size of dictionary = (number of filters, size of imagelets) = (44
1, 324)
average of filters =  -4.089243933727358e-06 +/- 0.001241096006797
0878
average energy of filters =  1.0 +/- 3.9562611248144994e-17
----------          HEH          ----------
size of dictionary = (number of filters, size of imagelets) = (44
1, 324)
average of filters =  -6.6112572753952305e-06 +/- 0.00121065448870
92556
average energy of filters =  1.0 +/- 3.700743415417188e-17
----------          EMP          ----------
size of dictionary = (number of filters, size of imagelets) = (44
1, 324)
average of filters =  4.993730484951632e-05 +/- 0.0012218228270885
788
average energy of filters =  1.0 +/- 3.700743415417188e-17
----------          HAP          ----------
size of dictionary = (number of filters, size of imagelets) = (44
1, 324)
average of filters =  -2.429586935952582e-05 +/- 0.001195729444507
5826
average energy of filters =  1.0 +/- 3.775513461943296e-17
```

## panel A: plotting some dictionaries

```
In [38]: pname = '/tmp/panel_A' #pname = fname + '_A'
```

```
In [39]: subplotpars = dict( left=0.042, right=1., bottom=0., top=1., wspace
         =0.05, hspace=0.05,)
         fig, axs = plt.subplots(3, 1, figsize=(fig_width/2, fig_width/(1+ph
         i)), gridspec_kw=subplotpars)

         for ax, color, homeo_method in zip(axs.ravel(), colors[1:], homeo_m
         ethods[1:]):
             ax.axis(c=color, lw=2, axisbg='w')
             ax.set_facecolor('w')
             from shl_scripts import show_dico
             fig, ax = show_dico(shl, dico[one_cv][homeo_method], data=data,
         dim_graph=dim_graph, fig=fig, ax=ax)
             # ax.set_ylabel(homeo_method)
             ax.text(-8, 7*dim_graph[0], homeo_method, fontsize=12, color=co
         lor, rotation=90)#, backgroundcolor='white'

         for ext in FORMATS: fig.savefig(pname + ext, dpi=dpi_export)
```

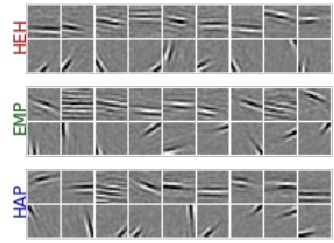

### panel B: quantitative comparison

```
In [40]: pname = '/tmp/panel_B' #fname + '_B'
```

```
In [41]:  from shl_scripts import time_plot
          variable = 'F'
          alpha = .3
          subplotpars = dict(left=0.2, right=.95, bottom=0.2, top=.95)#, wspa
          ce=0.05, hspace=0.05,)
          fig, ax = plt.subplots(1, 1, figsize=(fig_width/2, fig_width/(1+phi
          )), gridspec_kw=subplotpars)
          for i_cv in range(N_cv):
              for color, homeo_method in zip(colors, homeo_methods):
                  ax.axis(c='b', lw=2, axisbg='w')
                  ax.set_facecolor('w')
                  if i_cv==0:
                      fig, ax = time_plot(shl, dico[i_cv][homeo_method], vari
          able=variable, unit='bits', color=color, label=homeo_method, alpha=
          alpha_0, fig=fig, ax=ax)
                  else:
                      fig, ax = time_plot(shl, dico[i_cv][homeo_method], vari
          able=variable, unit='bits', color=color, alpha=alpha, fig=fig, ax=a
          x)
          ax.legend(loc='best')
          for ext in FORMATS: fig.savefig(pname + ext, dpi=dpi_export)
          if DEBUG: Image(pname +'.png')
```

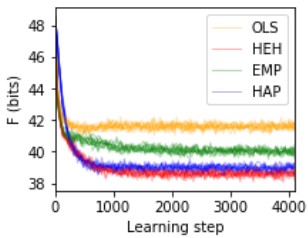

```
In [42]:  if DEBUG: Image(pname +'.png')
```

## Montage of the subplots

```
In [43]:  %%tikz -f pdf --save {fname}.pdf
          \draw[white, fill=white] (0.\linewidth,0) rectangle (1.\linewidth,
          .382\linewidth) ;
          \draw [anchor=north west] (.0\linewidth, .382\linewidth) node {\inc
          ludegraphics[width=.5\linewidth]{/tmp/panel_A}};
          \draw [anchor=north west] (.5\linewidth, .382\linewidth) node {\inc
          ludegraphics[width=.5\linewidth]{/tmp/panel_B}};
          \begin{scope}[font=\bf\sffamily\large]
          \draw [anchor=west,fill=white] (.0\linewidth, .382\linewidth) node
          [above right=-3mm] {$\mathsf{A}$};
          \draw [anchor=west,fill=white] (.53\linewidth, .382\linewidth) node
          [above right=-3mm] {$\mathsf{B}$};
          \end{scope}
```

```
!convert   -density {dpi_export} {fname}.pdf {fname}.jpg
!convert   -density {dpi_export} {fname}.pdf {fname}.png
#!convert   -density {dpi_export} -resize 5400  -units pixelsperinch
-flatten  -compress lzw  -depth 8 {fname}.pdf {fname}.tiff
Image(fname +'.png')
```

Out[44]:

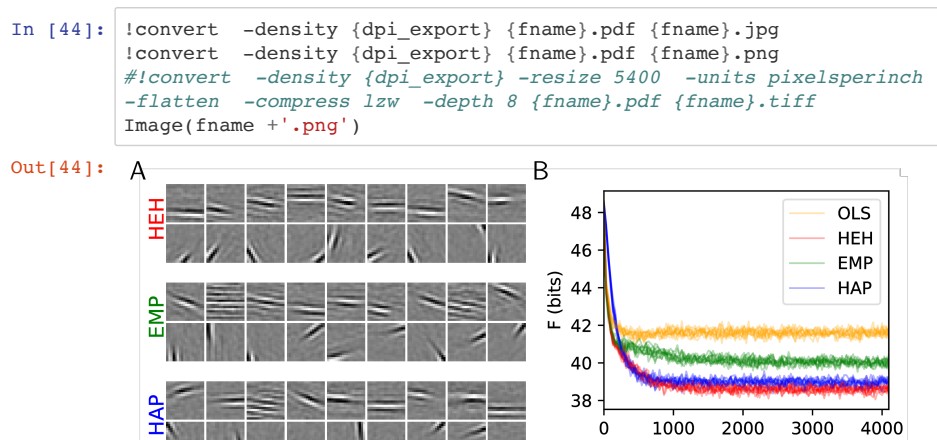

!echo "width=" ; convert {fname}.tiff -format "%[fx:w]" info: !echo ", \nheight=" ; convert {fname}.tiff -format "%[fx:h]" info: !echo ", \nunit=" ; convert {fname}.tiff -format "%U" info:!identify {fname}.tiff

## figure 4: Convolutional Neural Network

In [45]:
```
fname = 'figure_CNN'
```

```
In [46]: from CHAMP.DataLoader import LoadData
         from CHAMP.DataTools import LocalContrastNormalization, FilterInput
         Data, GenerateMask
         from CHAMP.Monitor import DisplayDico, DisplayConvergenceCHAMP, Dis
         playWhere

         import os
         datapath = os.path.join("/tmp", "database")
         path = os.path.join(datapath, "Raw_DataBase")
         TrSet, TeSet = LoadData('Face', path, decorrelate=False, resize=(65
         , 65))

         # MP Parameters
         nb_dico = 20
         width = 9
         dico_size = (width, width)
         l0 = 20
         seed = 42
         # Learning Parameters
         eta = .05
         nb_epoch = 500

         TrSet, TeSet = LoadData('Face', path, decorrelate=False, resize=(65
         , 65))
         N_TrSet, _, _, _ = LocalContrastNormalization(TrSet)
         Filtered_L_TrSet = FilterInputData(
             N_TrSet, sigma=0.25, style='Custom', start_R=15)

         mask = GenerateMask(full_size=(nb_dico, 1, width, width), sigma=0.8
         , style='Gaussian')

         from CHAMP.CHAMP_Layer import CHAMP_Layer

         from CHAMP.DataTools import SaveNetwork, LoadNetwork
         homeo_methods = ['None', 'HAP']

         for homeo_method, eta_homeo in zip(homeo_methods, [0., 0.0025]):
             ffname = 'cache_dir_CNN/CHAMP_low_' + homeo_method + '.pkl'
             try:
                 L1_mask = LoadNetwork(loading_path=ffname)
             except:
                 L1_mask = CHAMP_Layer(l0_sparseness=l0, nb_dico=nb_dico,
                                 dico_size=dico_size, mask=mask, verbose=1
         )
                 dico_mask = L1_mask.TrainLayer(
                     Filtered_L_TrSet, eta=eta, eta_homeo=eta_homeo, nb_epoc
         h=nb_epoch, seed=seed)
                 SaveNetwork(Network=L1_mask, saving_path=ffname)
```

**panel A: plotting some dictionaries**

```
In [47]: pname = '/tmp/panel_A' #pname = fname + '_A'
```

subplotpars = dict( left=0.042, right=1., bottom=0., top=1., wspace=0.05, hspace=0.05,) fig, axs =

plt.subplots(2, 1, figsize=(fig_width/2, fig_width/(1+phi)), gridspec_kw=subplotpars) for ax, color, homeo_method in zip(axs.ravel(), ['black', 'green'], homeo_methods): ax.axis(c=color, lw=2, axisbg='w') ax.set_facecolor('w') ffname = 'cache_dir/CHAMP_low_' + homeo_method + '.pkl' L1_mask = LoadNetwork(loading_path=ffname) fig, ax = DisplayDico(L1_mask.dictionary, fig=fig, ax=ax) # ax.set_ylabel(homeo_method) ax.text(-8, 7*dim_graph[0], homeo_method, fontsize=12, color=color, rotation=90)#, backgroundcolor='white' for ext in FORMATS: fig.savefig(pname + ext, dpi=dpi_export)

```
In [48]:  subplotpars = dict(left=0.042, right=1., bottom=0., top=1., wspace=
          0.05, hspace=0.05,)

          for color, homeo_method in zip(['black', 'green'], homeo_methods):
              #fig, axs = plt.subplots(1, 1, figsize=(fig_width/2, fig_width/
          (1+phi)), gridspec_kw=subplotpars)
              ffname = 'cache_dir_CNN/CHAMP_low_' + homeo_method + '.pkl'
              L1_mask = LoadNetwork(loading_path=ffname)
              fig, ax = DisplayDico(L1_mask.dictionary)
              # ax.set_ylabel(homeo_method)
              #for ax in list(axs):
              #    ax.axis(c=color, lw=2, axisbg='w')
              #    ax.set_facecolor('w')
              ax[0].text(-4, 3, homeo_method, fontsize=8, color=color, rotati
          on=90)#, backgroundcolor='white'
              plt.tight_layout( pad=0., w_pad=0., h_pad=.0)

              for ext in FORMATS: fig.savefig(pname + '_' + homeo_method + ex
          t, dpi=dpi_export)
```

<Figure size 576x28.8 with 0 Axes>

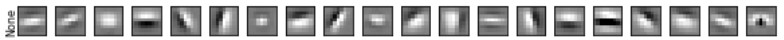

<Figure size 576x28.8 with 0 Axes>

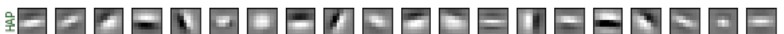

### panel B: quantitative comparison

```
In [49]:  pname = '/tmp/panel_B' #fname + '_B'
```

from shl_scripts import time_plot variable = 'F' alpha = .3 subplotpars = dict(left=0.2, right=.95, bottom=0.2, top=.95)#, wspace=0.05, hspace=0.05,) fig, axs = plt.subplots(2, 1, figsize=(fig_width/2, fig_width/(1+phi)), gridspec_kw=subplotpars) for ax, color, homeo_method in zip(axs, ['black', 'green'], homeo_methods): print(ax, axs) ffname = 'cache_dir_CNN/CHAMP_low_' + homeo_method + '.pkl' L1_mask = LoadNetwork(loading_path=ffname) fig, ax = DisplayConvergenceCHAMP(L1_mask, to_display=['histo'], fig=fig, ax=ax) ax.axis(c=color, lw=2, axisbg='w') ax.set_facecolor('w') # ax.set_ylabel(homeo_method) #ax.text(-8, 7*dim_graph[0], homeo_method, fontsize=12, color=color, rotation=90)#, backgroundcolor='white' for ext in FORMATS: fig.savefig(pname + ext, dpi=dpi_export) if DEBUG: Image(pname +'.png')

```
In [50]:  from shl_scripts import time_plot
          variable = 'F'
          alpha = .3
          subplotpars = dict(left=0.2, right=.95, bottom=0.2, top=.95)#, wspa
          ce=0.05, hspace=0.05,)

          for color, homeo_method in zip(['black', 'green'], homeo_methods):
              #fig, axs = plt.subplots(1, 1, figsize=(fig_width/2, fig_width/
          (1+phi)), gridspec_kw=subplotpars)
              ffname = 'cache_dir_CNN/CHAMP_low_' + homeo_method + '.pkl'
              L1_mask = LoadNetwork(loading_path=ffname)
              fig, ax = DisplayConvergenceCHAMP(L1_mask, to_display=['histo']
          , color=color)
              ax.axis(c=color, lw=2, axisbg='w')
              ax.set_facecolor('w')
              ax.set_ylabel('counts')
              ax.set_xlabel('feature #')
              ax.set_ylim(0, 560)
              #ax.text(-8, 7*dim_graph[0], homeo_method, fontsize=12, color=c
          olor, rotation=90)#, backgroundcolor='white'
              #ax[0].text(-8, 3, homeo_method, fontsize=12, color=color, rota
          tion=90)#, backgroundcolor='white'

              for ext in FORMATS: fig.savefig(pname + '_' + homeo_method + ex
          t, dpi=dpi_export)
              if DEBUG: Image(pname +'.png')
```

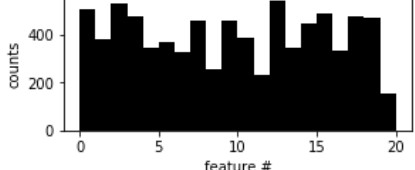

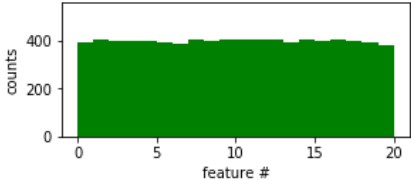

**Montage of the subplots**

```
In [51]: %ls -ltr /tmp/panel_*

         -rw-r--r--  1 501   wheel     67281 Nov 27 00:04 /tmp/panel_A.pdf
         -rw-r--r--  1 501   wheel     79492 Nov 27 00:04 /tmp/panel_A.png
         -rw-r--r--  1 501   wheel     49220 Nov 27 00:04 /tmp/panel_B.pdf
         -rw-r--r--  1 501   wheel    555716 Nov 27 00:04 /tmp/panel_B.png
         -rw-r--r--  1 501   wheel     27370 Nov 27 00:05 /tmp/panel_A_None.pd
         f
         -rw-r--r--  1 501   wheel     18909 Nov 27 00:05 /tmp/panel_A_None.pn
         g
         -rw-r--r--  1 501   wheel     26909 Nov 27 00:05 /tmp/panel_A_HAP.pdf
         -rw-r--r--  1 501   wheel     16431 Nov 27 00:05 /tmp/panel_A_HAP.png
         -rw-r--r--  1 501   wheel      8816 Nov 27 00:05 /tmp/panel_B_None.pd
         f
         -rw-r--r--  1 501   wheel     39035 Nov 27 00:05 /tmp/panel_B_None.pn
         g
         -rw-r--r--  1 501   wheel      8813 Nov 27 00:05 /tmp/panel_B_HAP.pdf
         -rw-r--r--  1 501   wheel     38743 Nov 27 00:05 /tmp/panel_B_HAP.png
```

```
In [52]: fname
```

```
Out[52]: 'figure_CNN'
```

```
In [53]: 382+191
```

```
Out[53]: 573
```

```
In [54]: %%tikz -f pdf --save {fname}.pdf
         \draw[white, fill=white] (0.\linewidth,0) rectangle (1.\linewidth,
         .382\linewidth) ;
         \draw [anchor=north west] (.0\linewidth, .375\linewidth) node {\inc
         ludegraphics[width=.95\linewidth]{/tmp/panel_A_None}};
         \draw [anchor=north west] (.0\linewidth, .300\linewidth) node {\inc
         ludegraphics[width=.95\linewidth]{/tmp/panel_A_HAP}};
         \draw [anchor=north west] (.0\linewidth, .191\linewidth) node {\inc
         ludegraphics[width=.45\linewidth]{/tmp/panel_B_None}};
         \draw [anchor=north west] (.5\linewidth, .191\linewidth) node {\inc
         ludegraphics[width=.45\linewidth]{/tmp/panel_B_HAP}};
         \begin{scope}[font=\bf\sffamily\large]
         %\draw [anchor=west,fill=white] (.0\linewidth, .382\linewidth) node
         [above right=-3mm] {$\mathsf{A}$};
         \draw [anchor=west,fill=white] (.0\linewidth, .191\linewidth) node
         [above right=-3mm] {$\mathsf{A}$};
         \draw [anchor=west,fill=white] (.53\linewidth, .191\linewidth) node
         [above right=-3mm] {$\mathsf{B}$};
         \end{scope}
```

```
In [55]:  !convert  -density {dpi_export} {fname}.pdf {fname}.jpg
          !convert  -density {dpi_export} {fname}.pdf {fname}.png
          #!convert  -density {dpi_export} -resize 5400  -units pixelsperinch
          -flatten -compress lzw  -depth 8 {fname}.pdf {fname}.tiff
          Image(fname +'.png')
```

Out[55]:

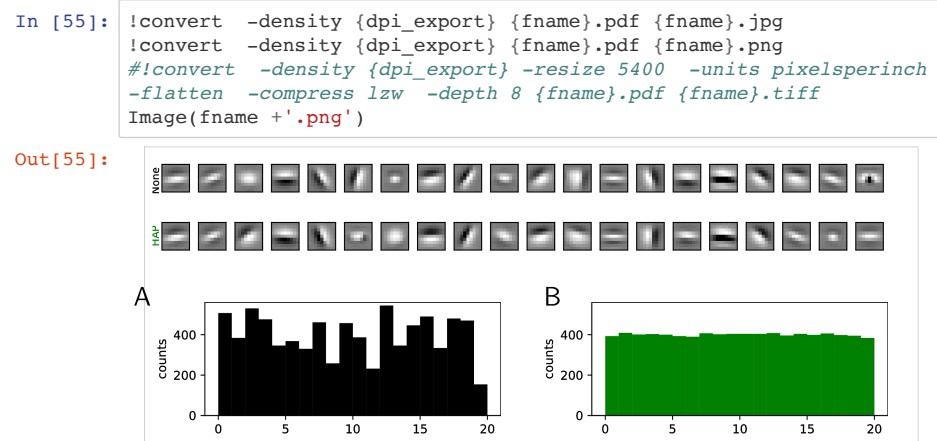

!echo "width=" ; convert {fname}.tiff -format "%[fx:w]" info: !echo ", \nheight=" ; convert {fname}.tiff -format
"%[fx:h]" info: !echo ", \nunit=" ; convert {fname}.tiff -format "%U" info:!identify {fname}.tiff

## coding

The learning itself is done via a gradient descent but is highly dependent on the coding / decoding
algorithm. This belongs to a another function (in the shl_encode.py
(https://github.com/bicv/SHL_scripts/blob/master/shl_scripts/shl_encode.py) script)

# Supplementary controls

## starting a learning

```
In [ ]:  shl = SHL(**opts)
         list_figures = ['show_dico', 'show_Pcum', 'time_plot_F']
         dico = shl.learn_dico(data=data, list_figures=list_figures, matname
         =tag + '_vanilla')
```

```
In [ ]:  print('size of dictionary = (number of filters, size of imagelets)
         = ', dico.dictionary.shape)
         print('average of filters = ',  dico.dictionary.mean(axis=1).mean()
         ,
                '+/-',  dico.dictionary.mean(axis=1).std())
         SE = np.sqrt(np.sum(dico.dictionary**2, axis=1))
         print('average energy of filters = ', SE.mean(), '+/-', SE.std())
```

## getting help

```
In [ ]: help(shl)
```

```
In [ ]: help(dico)
```

## loading a database

Loading patches, with or without mask:

```
In [ ]: N_patches = 12
        from shl_scripts.shl_tools import show_data
        opts_ = opts.copy()
        opts_.update(verbose=0)
        for i, (do_mask, label) in enumerate(zip([False, True], ['Without m
        ask', 'With mask'])):
            data_ = SHL(DEBUG_DOWNSCALE=1, N_patches=N_patches, n_image=1,
        do_mask=do_mask, seed=seed, **opts_).get_data()
            fig, axs = show_data(data_)
            axs[0].set_ylabel(label);
            plt.show()
```

## Testing different algorithms

```
In [ ]: fig, ax = None, None

        for homeo_method in ['None', 'HAP']:
            for algorithm in ['lasso_lars', 'lars', 'elastic', 'omp', 'mp']
        : # 'threshold', 'lasso_cd',
                opts_ = opts.copy()
                opts_.update(homeo_method=homeo_method, learning_algorithm=
        algorithm, verbose=0)
                shl = SHL(**opts_)
                dico= shl.learn_dico(data=data, list_figures=[],
                                matname=tag + ' - algorithm={}'.format(algor
        ithm) + ' - homeo_method={}'.format(homeo_method))
                fig, ax = shl.time_plot(dico, variable='F', fig=fig, ax=ax,
        label=algorithm +'_' + homeo_method)

            ax.legend()
```

## Testing two different dictionary initalization strategies

White Noise Initialization + Learning

```
In [ ]: shl = SHL(one_over_F=False, **opts)
        dico_w = shl.learn_dico(data=data, matname=tag + '_WHITE', list_fig
        ures=[])
        shl = SHL(one_over_F=True, **opts)
        dico_1oF = shl.learn_dico(data=data, matname=tag + '_OVF', list_fig
        ures=[])
        fig_error, ax_error = None, None
        fig_error, ax_error = shl.time_plot(dico_w, variable='F', fig=fig_e
        rror, ax=ax_error, color='blue', label='white noise')
        fig_error, ax_error = shl.time_plot(dico_1oF, variable='F', fig=fig
        _error, ax=ax_error, color='red', label='one over f')
        #ax_error.set_ylim((0, .65))
        ax_error.legend(loc='best')
```

## Testing two different learning rates strategies

We use by defaut the strategy of ADAM, see [https://arxiv.org/pdf/1412.6980.pdf](https://arxiv.org/pdf/1412.6980.pdf)
[(https://arxiv.org/pdf/1412.6980.pdf)](https://arxiv.org/pdf/1412.6980.pdf)

```
In [ ]: shl = SHL(beta1=0., **opts)
        dico_fixed = shl.learn_dico(data=data, matname=tag + '_fixed', list
        _figures=[])
        shl = SHL(**opts)
        dico_default = shl.learn_dico(data=data, matname=tag + '_default',
        list_figures=[])
        fig_error, ax_error = None, None
        fig_error, ax_error = shl.time_plot(dico_fixed, variable='F', fig=f
        ig_error, ax=ax_error, color='blue', label='fixed')
        fig_error, ax_error = shl.time_plot(dico_default, variable='F', fig
        =fig_error, ax=ax_error, color='red', label='ADAM')
        #ax_error.set_ylim((0, .65))
        ax_error.legend(loc='best')
```

## Testing different number of neurons and sparsity

As suggested by AnonReviewer3, we have tested how the convergence was modified by changing the number of neurons. By comparing different numbers of neurons we could re-draw the same figures for the convergence of the algorithm as in our original figures. In addition, we have also checked that this result will hold on a range of sparsity levels. In particular, we found that in general, increasing the `l0_sparseness` parameter, the convergence took progressively longer. Importantly, we could see that in both cases, this did not depend on the kind of homeostasis heuristic chosen, proving the generality of our results.

This is shown in the supplementary material that we have added to our revision ("Testing different number of neurons and sparsity") . This useful extension proves the originality of our work as highlighted in point 4, and the generality of these results compared to the parameters of the network.

```
In [ ]: from shl_scripts.shl_experiments import SHL_set
        homeo_methods = ['None', 'OLS', 'HEH']
        homeo_methods = ['None', 'EMP', 'HAP', 'HEH', 'OLS']

        variables = ['l0_sparseness', 'n_dictionary']
        list_figures = []

        #n_dictionary=21**2

        for homeo_method in homeo_methods:
            opts_ = opts.copy()
            opts_.update(homeo_method=homeo_method, datapath=datapath)
            experiments = SHL_set(opts_, tag=tag + '_' + homeo_method)
            experiments.run(variables=variables, n_jobs=1, verbose=0)

        fig, axs = plt.subplots(len(variables), 1, figsize=(fig_width/2, fi
        g_width/(1+phi)), gridspec_kw=subplotpars, sharey=True)

        for i_ax, variable in enumerate(variables):
            for color, homeo_method in zip(colors, homeo_methods):
                opts_ = opts.copy()
                opts_.update(homeo_method=homeo_method, datapath=datapath)
                experiments = SHL_set(opts_, tag=tag + '_' + homeo_method)
                fig, axs[i_ax] = experiments.scan(variable=variable, list_f
        igures=[], display='final', fig=fig, ax=axs[i_ax], color=color, dis
        play_variable='F', verbose=0) #, label=homeo_metho
                axs[i_ax].set_xlabel('') #variable
                axs[i_ax].text(.1, .8,  variable, transform=axs[i_ax].trans
        Axes)
                #axs[i_ax].get_xaxis().set_major_formatter(matplotlib.ticke
        r.ScalarFormatter())
```

# Perspectives

# Convolutional neural networks

```
In [ ]: from CHAMP.DataLoader import LoadData
        from CHAMP.DataTools import LocalContrastNormalization, FilterInput
        Data, GenerateMask
        from CHAMP.Monitor import DisplayDico, DisplayConvergenceCHAMP, Dis
        playWhere

        import os
        home = os.getenv('HOME')
        datapath = os.path.join("/tmp", "database")
        path = os.path.join(datapath, "Face_DataBase")
        TrSet, TeSet = LoadData('Face', path, decorrelate=False, resize=(65
        , 65))
        to_display = TrSet[0][0, 0:10, :, :, :]
        print('Size=', TrSet[0].shape)
        DisplayDico(to_display)
```

**Training on a face database**

```
In [ ]: # MP Parameters
        nb_dico = 20
        width = 9
        dico_size = (width, width)
        l0 = 20
        seed = 42
        # Learning Parameters
        eta = .05
        nb_epoch = 500

        TrSet, TeSet = LoadData('Face', path, decorrelate=False, resize=(65
        , 65))
        N_TrSet, _, _, _ = LocalContrastNormalization(TrSet)
        Filtered_L_TrSet = FilterInputData(
            N_TrSet, sigma=0.25, style='Custom', start_R=15)
        to_display = Filtered_L_TrSet[0][0, 0:10, :, :, :]
        DisplayDico(to_display)

        mask = GenerateMask(full_size=(nb_dico, 1, width, width), sigma=0.8
        , style='Gaussian')
        DisplayDico(mask)
```

**Training the ConvMP Layer with homeostasis**

```
In [ ]:  from CHAMP.CHAMP_Layer import CHAMP_Layer

         from CHAMP.DataTools import SaveNetwork, LoadNetwork
         fname = 'cache_dir_CNN/CHAMP_low_None.pkl'
         try:
             L1_mask = LoadNetwork(loading_path=fname)
         except:
             L1_mask = CHAMP_Layer(l0_sparseness=l0, nb_dico=nb_dico,
                             dico_size=dico_size, mask=mask, verbose=2)
             dico_mask = L1_mask.TrainLayer(
                 Filtered_L_TrSet, eta=eta, nb_epoch=nb_epoch, seed=seed)
             SaveNetwork(Network=L1_mask, saving_path=fname)

         DisplayDico(L1_mask.dictionary)
         DisplayConvergenceCHAMP(L1_mask, to_display=['error', 'histo'])
         DisplayWhere(L1_mask.where)
```

## Training the ConvMP Layer with homeostasis

```
In [ ]:  fname = 'cache_dir_CNN/CHAMP_low_HAP.pkl'
         try:
             L1_mask = LoadNetwork(loading_path=fname)
         except:

             # Learning Parameters
             eta_homeo = 0.0025
             L1_mask = CHAMP_Layer(l0_sparseness=l0, nb_dico=nb_dico,
                                 dico_size=dico_size, mask=mask, verbose=1
         )
             dico_mask = L1_mask.TrainLayer(
                 Filtered_L_TrSet, eta=eta, eta_homeo=eta_homeo, nb_epoch=nb
         _epoch, seed=seed)
             SaveNetwork(Network=L1_mask, saving_path=fname)

         DisplayDico(L1_mask.dictionary)
         DisplayConvergenceCHAMP(L1_mask, to_display=['error', 'histo'])
         DisplayWhere(L1_mask.where)
```

## Reconstructing the input image

```
In [ ]:  from CHAMP.DataTools import Rebuilt
         import torch
         rebuilt_image = Rebuilt(torch.FloatTensor(L1_mask.code), L1_mask.di
         ctionary)
         DisplayDico(rebuilt_image[0:10, :, :, :])
```

## Training the ConvMP Layer with higher-level filters

We train higher-level feature vectors by forcing the network to :

- learn bigger filters,
- represent the information using a bigger dictionary (higher sparseness)
- represent the information with less features (higher sparseness)

```
In [ ]: fname = 'cache_dir_CNN/CHAMP_high_None.pkl'
        try:
            L1_mask = LoadNetwork(loading_path=fname)
        except:

            nb_dico = 60
            width = 19
            dico_size = (width, width)
            l0 = 5
            mask = GenerateMask(full_size=(nb_dico, 1, width, width), sigma
        =0.8, style='Gaussian')
            # Learning Parameters
            eta_homeo = 0.0
            eta = .05
            nb_epoch = 500
            # learn
            L1_mask = CHAMP_Layer(l0_sparseness=l0, nb_dico=nb_dico,
                                  dico_size=dico_size, mask=mask, verbose=0
        )
            dico_mask = L1_mask.TrainLayer(
                Filtered_L_TrSet, eta=eta, eta_homeo=eta_homeo, nb_epoch=nb
        _epoch, seed=seed)
            SaveNetwork(Network=L1_mask, saving_path=fname)

        DisplayDico(L1_mask.dictionary)
        DisplayConvergenceCHAMP(L1_mask, to_display=['error'])#, 'histo'])
        DisplayWhere(L1_mask.where)
```

```
In [ ]: fname = 'cache_dir_CNN/CHAMP_high_HAP.pkl'
        try:
            L1_mask = LoadNetwork(loading_path=fname)
        except:

            nb_dico = 60
            width = 19
            dico_size = (width, width)
            l0 = 5
            mask = GenerateMask(full_size=(nb_dico, 1, width, width), sigma
        =0.8, style='Gaussian')
            # Learning Parameters
            eta_homeo = 0.0025
            eta = .05
            nb_epoch = 500
            # learn
            L1_mask = CHAMP_Layer(l0_sparseness=l0, nb_dico=nb_dico,
                                 dico_size=dico_size, mask=mask, verbose=0
        )
            dico_mask = L1_mask.TrainLayer(
                Filtered_L_TrSet, eta=eta, eta_homeo=eta_homeo, nb_epoch=nb
        _epoch, seed=seed)
            SaveNetwork(Network=L1_mask, saving_path=fname)

        DisplayDico(L1_mask.dictionary)
        DisplayConvergenceCHAMP(L1_mask, to_display=['error'])#, 'histo'])
        DisplayWhere(L1_mask.where)
```

### Training on MNIST database

fname = 'cache_dir_CNN/CHAMP_MNIST_HAP.pkl' try: L1_mask = LoadNetwork(loading_path=fname)
except: path = os.path.join(datapath, "MNISTtorch") TrSet, TeSet = LoadData('MNIST', data_path=path)
N_TrSet, _, _, _ = LocalContrastNormalization(TrSet) Filtered_L_TrSet = FilterInputData( N_TrSet, sigma=0.25,
style='Custom', start_R=15) nb_dico = 60 width = 7 dico_size = (width, width) l0 = 15 # Learning Parameters
eta_homeo = 0.0025 eta = .05 nb_epoch = 500 # learn L1_mask = CHAMP_Layer(l0_sparseness=l0,
nb_dico=nb_dico, dico_size=dico_size, mask=mask, verbose=2) dico_mask = L1_mask.TrainLayer(
Filtered_L_TrSet, eta=eta, eta_homeo=eta_homeo, nb_epoch=nb_epoch, seed=seed)
SaveNetwork(Network=L1_mask, saving_path=fname) DisplayDico(L1_mask.dictionary)
DisplayConvergenceCHAMP(L1_mask, to_display=['error', 'histo']) DisplayWhere(L1_mask.where)

# Computational details

## caching simulation data

A convenience script to run and cache most learning items in this notebooks:

```
In [ ]:  !ls -l {shl.cache_dir}/{tag}*
         #!rm {shl.cache_dir}/{tag}*lock*
         #!rm {shl.cache_dir}/{tag}*
         #!ls -l {shl.cache_dir}/{tag}*
```

```
In [ ]:  %%writefile model.py
         #!/usr/bin/env python3
         # -*- coding: utf-8 -*
         tag = 'ICLR'
         from shl_scripts.shl_experiments import SHL, prun
         # pre-loading data
         datapath = '../../SparseHebbianLearning/database'
         # different runs
         #opts = dict(cache_dir='cache_dir_ICLR', datapath=datapath, verbose
         =0)
         #opts = dict(cache_dir='cache_dir_cluster', datapath=datapath, verb
         ose=0)
         opts = dict(datapath=datapath, verbose=0)

         shl = SHL(**opts)
         data = shl.get_data(matname=tag)

         # running main simulations
         # Figure 1 & 3
         N_cv = 10
         homeo_methods = ['None', 'OLS', 'HEH', 'HAP', 'EMP']
         seed = 42

         # running in parallel on a multi-core machine
         import sys
         try:
             n_jobs = int(sys.argv[1])
             print('n_jobs=', n_jobs)
         except:
             n_jobs = 4
             n_jobs = 9
             n_jobs = 10
             n_jobs = 1
             n_jobs = 35

         if n_jobs>0:

             list_figures = []

             from shl_scripts.shl_experiments import SHL_set
             for homeo_method in homeo_methods:
                 opts_ = opts.copy()
                 opts_.update(homeo_method=homeo_method)
                 experiments = SHL_set(opts_, tag=tag + '_' + homeo_method,
         N_scan=N_cv)
                 experiments.run(variables=['seed'], n_jobs=n_jobs, verbose=
         0)

             # Figure 2-B
             variables = ['eta', 'alpha_homeo', 'eta_homeo']
```

```python
    variables = ['eta', 'eta_homeo', 'l0_sparseness', 'n_dictionary
']
    bases = [10, 10, 2, 2]

    for homeo_method, base in zip(homeo_methods, bases):
        opts_ = opts.copy()
        opts_.update(homeo_method=homeo_method)
        experiments = SHL_set(opts_, tag=tag + '_' + homeo_method,
base=base)
        experiments.run(variables=variables, n_jobs=n_jobs, verbose
=0)

    # Annex X.X

    shl = SHL(**opts)
    dico = shl.learn_dico(data=data, list_figures=list_figures, mat
name=tag + '_vanilla')

    for algorithm in ['lasso_lars', 'lasso_cd', 'lars', 'elastic',
'omp', 'mp']: # 'threshold',
        opts_ = opts.copy()
        opts_.update(homeo_method='None', learning_algorithm=algori
thm, verbose=0)
        shl = SHL(**opts_)
        dico= shl.learn_dico(data=data, list_figures=[],
                        matname=tag + ' - algorithm={}'.format(algor
ithm))

    for homeo_method in ['None', 'HAP']:
        for algorithm in ['lasso_lars', 'lars', 'elastic', 'omp', '
mp']: # 'threshold', 'lasso_cd',
            opts_ = opts.copy()
            opts_.update(homeo_method=homeo_method, learning_algori
thm=algorithm, verbose=0)
            shl = SHL(**opts_)
            dico= shl.learn_dico(data=data, list_figures=[],
                        matname=tag + ' - algorithm={}'.format(a
lgorithm) + ' - homeo_method={}'.format(homeo_method))

    shl = SHL(one_over_F=False, **opts)
    dico_w = shl.learn_dico(data=data, matname=tag + '_WHITE', list
_figures=[])
    shl = SHL(one_over_F=True, **opts)
    dico_1oF = shl.learn_dico(data=data, matname=tag + '_OVF', list
_figures=[])

    shl = SHL(beta1=0., **opts)
    dico_fixed = shl.learn_dico(data=data, matname=tag + '_fixed',
list_figures=[])
    shl = SHL(**opts)
    dico_default = shl.learn_dico(data=data, matname=tag + '_defaul
t', list_figures=[])
```

```python
In [ ]: %run model.py 0
```

## Version used

```
In [ ]: %load_ext version_information
        %version_information numpy, shl_scripts
```

## version control

```
In [ ]: !git status
```

```
In [ ]: !git pull
```

```
In [ ]: !git commit -am' {tag} : re-running notebooks'
```

```
In [ ]: !git push
```

## exporting the notebook

```
In [ ]: !jupyter nbconvert Annex.ipynb
```

```
In [ ]: #!jupyter-nbconvert --template report --to pdf Annex.ipynb
```

```
In [ ]: !pandoc Annex.html -o Annex.pdf
```

```
In [ ]: !zip Annex.zip Annex.html
```

Done. Thanks for your attention!