# OpenReview forum: "An adaptive homeostatic algorithm for the unsupervised learning of visual features"
_ICLR.cc/2019/Conference_

### Official Review · AnonReviewer1 · 2018-10-29
**Well written paper, with good literature review and interesting experiments showing faster unsupervised learning**

**Rating:** 9
**Confidence:** 4

**Review:**

This paper proposed a bio-inspired sparse coding algorithm where iterations
for dictionary updates take into account the past updates. It is argued
that time takes a crucial rule in learning.

The paper is quite well written and contains an extensive literature review
demonstrating a good understanding of previous literature in both ML/DL and biological
vision.

The idea of using a "non-linear gain normalization" to adjust atom selection
in sparse coding is interesting and as far as I know novel, while providing
interesting empirical results: The system learns in an unsupervised way faster.

Misc:

- Using < > for latex brakets is not ideal. I would recommend: $\langle\,,\rangle$

- "derivable" I guess you mean "differentiable"

- Oliphant and Hunter are cited for Numpy/scipy and matplotlib but the
reference to Pedregosa et al. for sklearn is missing.

---

> ### Author Response · Authors · 2018-11-23
> **Response To AnonReviewer1**
>
> We thank AnonReviewer1 for the careful reading and encouraging comments.
>
> Indeed, the idea of a non-linear or adaptive gain normalization is novel to our knowledge, and is the main reason for deciding to submit this work to ICLR. We based our theoretical insight on an extensive reading and experience on neurophysiological data which we tried as much as possible to reconcile with the latest literature in ML/DL.
>
> In particular, we think that this problem is resolved in most DL approaches using heuristics such as dropout ( http://www.jmlr.org/papers/volume15/srivastava14a/srivastava14a.pdf ) or batch normalization ( https://arxiv.org/abs/1502.03167 ). We acknowledge that the objective we use (equiprobability) may seem arbitrary, but we think that 1) it best fits constraints in biological populations of neurons 2) it can be adapted to other priors on the desired probability of nodes in the network.
>
> In our current revision,  while keeping the same theoretical framework and simulation results, we have highlighted our main contributions: 1/ to show that $\ell_2$ normalization leads to non-homogeneous data 2/ provide with an exact rule 3/ propose a simplfied heurstics and show its effectiveness.
>
> Also, we have fixed the typos and minor issues  (in "Misc") in the revision that is being uploaded to the openreview preprint server.
>
> Thanks again for your careful reading.

---

### Official Review · AnonReviewer3 · 2018-11-03
**Solid work but the importance unclear**

**Rating:** 4
**Confidence:** 4

**Review:**

Please consider this rubric when writing your review:
1. Briefly establish your personal expertise in the field of the paper.
2. Concisely summarize the contributions of the paper.
3. Evaluate the quality and composition of the work.
4. Place the work in context of prior work, and evaluate this work's novelty.
5. Provide critique of each theorem or experiment that is relevant to your judgment of the paper's novelty and quality.
6. Provide a summary judgment if the work is significant and of interest to the community.

1. I am a researcher working at the intersection of machine learning,
computational neuroscience and biological vision.  I have experience
with neural network models and visual neurophysiology.

2. This paper develops and tests an adaptive homeostatic algorithm for
unsupervised visual feature learning (for example for learning models
of early visual processing/V1).

3.The work spends a lot of pages describing the general problem of
unsupervised feature learning and the history of the base algorithms.
The literature review is quite extensive.  The new content appears to
be in section 2.2 (Histogram Equalization Homeostasis - HEH), where a
simple idea to keep all units with balanced activity over the set of
natural images.  The authors also develop a computationally cheaper
version they call HAP (Homeostasis on Activation Probability) The
authors show that their F function is optimized quicker with the HEH
and HAP algorithms.  I would like to see how these curves vary with
the number of neurons (e.g. can you add X% more neurons and get
similar convergence speed -- and if so which is more computationally
costly)?

4. Many groups have developed various homeostatic algorithms for
unsupervised learning, though I have not seen this exact one before.

5.  The experiments reveal the resulting receptive fields and show the
decrease in the F function (error function).    The resulting receptive fields
do not seem that different to me between the different methods.  I am also not
that convinced that the faster convergence as a function of learning step is that important
especially as the learning steps may be more computationally expensive for this method.

6. I am not sure how interesting this work will be for the ICLR audience,
as it is not clear how important the faster convergence and more even
utilization of neurons is (and how it would compare computationally
with just having more neurons).

---

> ### Author Response · Authors · 2018-11-23
> **Response To AnonReviewer3**
>
> We thank the reviewer and its careful reading of our paper. Concerning point 6: Indeed, we acknowledge that this type of paper may be unconventional for the audience at ICLR. But we strongly  believe that scientific knowledge on biological vision is essential to work out the models that will shape DL in the future. Thus, we fully understand the rating given by the reviewer and would like to suggest that our revision addresses the main comment and show that it is relevant for a presentation at ICLR.
>
> First, we have extended the results by using the useful suggestions of AnonReviewer3 (point 3):
> As suggested by the reviewer we have tested how the convergence was modified by changing the number of neurons. By comparing different numbers of neurons we could re-draw the same figures for the convergence of the algorithm as in our original figures. In addition, we have also checked that this result will hold on a range of sparsity levels. In particular, we found that in general, increasing the l0_sparseness parameter, the convergence took progressively longer.  Importantly, we could see that in both cases, this did not depend on the kind of homeostasis heuristic chosen, proving the generality of our results.
>
> This is shown in the supplementary material that we have added to our revision (section "Testing different number of neurons and sparsity") . This useful extension proves the originality of our work as highlighted in point 4, and the generality of these results compared to the parameters of the network.
>
> Second, the comment made in point 5 is essential: figures 1 and 3  in our first revision where not showing appropriately the qualitative improvement which is achieved in the resulting filters. Indeed, we were showing 18 atoms chosen at random from the 441 filters from the dictionary. We initially thought that this "blind" shuffling would be a fair representation of the data, but as revealed by point 5, this was not true. We have now changed the strategy by now showing  "the upper and lower row respectively show the least and most probably selected atoms."  (see captions of figures 1 and 3). This now shows clearly the qualitative improvement in using a proper homeostasis and in particular that using the $\ell_2$ normalization leads to the emergence of filters which are aberrant (too or not enough selective). In particular, we now show quantitatively the probability of choice of each atom - showing that most active filters are used twice more as least active ones.
>
> Finally, we have made an extensive pass on the manuscript to take into account the different points  and make sure that this approach derived from biological vision is relevant for the audience at ICLR.
>
> As such, we believe this major change in the way we present the work, both in the quality of the resulting filters and in the generality of the results, have significantly changed the scope of our work to justify its acceptance to ICLR. We thank again the reviewer for these very useful contributions to our work.

---

### Official Review · AnonReviewer4 · 2018-11-16
**Well written but poorly motivated**

**Rating:** 5
**Confidence:** 5

**Review:**

This paper discusses the addition of a regularizer to a standard sparse coding/dictionary learning algorithm to encourage the atoms to be used with uniform frequency.    I do not think this work should be accepted to the conference for the following reasons:

1: The authors show no benefit of this scheme except perhaps faster convergence.  If faster training of dictionary learning models was a bottleneck in practical applications, this might be of interest, but it is not.  SPAMS (http://spams-devel.gforge.inria.fr/) can train a model on image patches as the authors do here in a few tens of seconds on a modern computer.  On the other hand, the authors give no evidence, empirical or otherwise, that their method is useful on any downstream tasks.
In my view, they do not even show that the distribution of atom usage will be better with their algorithm after the learning has converged, as at least according to their learning curves, the baselines have not finished converging.  It is not even clear that the final compression of the baselines would not be better.  Even if they did show these convincingly, it is not obvious to me that it is valuable; the authors need to *show* that uniform usage is desirable.

2:    The authors should compare against several costs/algorithms (e.g. l_0 with OMP, l_1 with LARS, etc.), and across various N_0/sparsity penalties, and across several datasets.   The empirical evaluation is quite weak- one sparsity setting, two baselines, one dataset.  Even without the "train to convergence" question above, I don't think the authors have demonstrated that their claims on the properties of their algorithms/formulations are generally true.

---

> ### Author Response · Authors · 2018-11-23
> **Response to AnonReviewer4**
>
> We thank the reviewer for having taken the time to judge our paper and to have detailed his judgement on their two points. We would like to point out that AnonReviewer4's final quantitative score as well as the confidence given will be crucial for the fact that this paper will or will not be presented at ICLR. We would like to respectfully detail how we completely disagree with the comments given in the two points, but acknowledge that this was mainly due to the way we presented the motivation for the paper.  We hope the revised version of the paper now meets the standards for ICLR and justifies to update the "red flag" (clear rejection) to a green light.
>
> First, the goal is not faster computation on a CPU. Our (github-shared) code runs in a few dozens of seconds per learning on a standard laptop - but the goal is mainly to be able to test all parameters. We have not used SPAMS in this work as we could use the similar methods which are used in the sklearn library. However, SPAMS is a great inspiration for our framework. (For information, the complete simulations for this paper take approximately 12 hours --which are easily distributed on a cluster as we multiplied the number of independent learning runs using different classes of parameters, cross-validations and types of sparse coding algorithms - in total approx 500 experiments. It takes a dozens of minutes on a 100 nodes cluster.). Our motivation is mainly to understand biological vision and hope this would percolate to ML. Yes, we obtain faster convergence, but as an epiphenomenon of the better efficiency of our adaptive homeostatic algorithm. However, we agree that this was not clear in this first revision: atoms which were displayed looked qualitatively similar. We have solved this issue thanks to the comments of the anonymous reviewers by now displaying the most and least active atoms. This shows a clear distinction between different methods and an important result: when $\ell_2$ normalizing atoms, dictionary learning may converge to a result for which the ratio of activity between the most activated and the least activated is of the order 2. This result is often overlooked in dictionary learning and is a first novel result of the paper.
>
> This being said, Figures 1 and 3 now show the clear qualitative advantage of using homeostasis in unsupervised learning. This now certainly allow to understand *why* convergence speed is a good indicator ---not for an advantage on the running speed on a classical CPU--- but rather in showing that this allows a more efficient dictionary learning overall. Concerning the point " It is not even clear that the final compression of the baselines would not be better.  Even if they did show these convincingly, it is not obvious to me that it is valuable.", we have performed the same experiments on more iterations such that we clearly see that baseline stay separate. Finally on the same point, we have not used at this point any application, such as supervised learning,  as it is out of the scope of this paper. But we thank the reviewer for suggesting it.
>
> Second, we had already done the comparison "against several costs/algorithms (e.g. l_0 with OMP, l_1 with LARS, etc.), and across various N_0/sparsity penalties" but we had initially omitted to include this supplementary data (that takes the form of a single jupyter notebook which allows to reproduce all results). We have now included it in an anonymized format. This supplementary material contains code to replicate all figures but also additional experiments to test the effect of the different parameters. In short, we verified that the results we present are valid over a various number of parameters of the network, like the learning rates (figure 2) but also sparsity and the size of the dictionary (see Response To AnonReviewer3 @ https://openreview.net/forum?id=SyMras0cFQ&noteId=BylQtQPHRX ). As in Sandin, 2017 paper we have shown similar results in OMP. We are in the process of extending this framework to other sparse coding algorithms (LARS and lasso_lars) as plugged in from sklearn without any modification (in theory) to these algorithms. Indeed, we should remind that our adaptive homeostasis allows to be implemented by modifying the norm of each atom of the dictionary (as was done in the original work by Olshausen). We also show in the paper the application to a one-layer convolution network and our preliminary results show that we can extend this to a hierarchical network.
>
> I hope that with these clarifications on the form we gave to the paper (without changing the theory behind it), the statement that " I don't think the authors have demonstrated that their claims on the properties of their algorithms/formulations are generally true."  could be re-assessed to allow us to share this work inspired by biology to the ICLR community.

---

> > ### Comment · AnonReviewer4 · 2018-11-29
> > **Additional experiments make the paper stronger, but it is still not well motivated**
> >
> > The authors in their revision have convinced me that their algorithm does what they say it does; and am happy to concede that "I don't think the authors have demonstrated that their claims on the properties of their algorithms/formulations are generally true." is no longer valid.
> >
> >    However,  I am still not convinced that the presented work is relevant or useful.  Statements like " ... Figures 1 and 3 now show the clear qualitative advantage of using homeostasis in unsupervised learning ..." are mystifying to me:  *why* is there advantage? The same with "may converge to a result for which the ratio of activity between the most activated and the least activated is of the order 2".  Why is this a problem?   What order would not be a problem?     The goal of unsupervised learning is rarely compression for its own sake (and even when this is the goal, measuring success in the space of images, for example,  requires human evaluation).  Furthermore, it is not unusual for feature representations that are optimally compressed to be less useful for other tasks.  Sparse coding gained popularity in the machine learning community because it lead to SOTA algorithms in image denoising, super-resolution,  and object recognition.   Does this approach improve the results on any of these?   Is the improvement enough to surpass modern approaches to these problems?  Is there some other downstream task where the authors method makes a significant difference?  The authors say that "This result is often overlooked in dictionary learning and is a first novel result of the paper. ".   My claim is that this result is not unknown; but rather, has not been generally discussed in the broader sparse coding literature because it has not been considered a serious problem.  On the other hand, in the clustering literature (note that clustering is a very particular form of l_0 sparse coding), where cluster balance can be a serious problem with downstream consequences,  there are many works investigating cluster balancing.
> >
> > I am changing my score to 5.

---

### Author Response · Authors · 2018-11-22
**Motivation and main points on the changes operated on the presentation of the results**

We appreciate the feedback from the reviewers, especially looking at the qualitative judgments ("solid work", "well written", "interesting experiments showing faster unsupervised learning"). However, we would like to highlight that the quantitative evaluation (9 / 4/  3) is not consistent. Basing selection on the mean score makes it highly improbable that, as it is, the paper could be presented at the audience of ICLR.

We believe that it is possible to fix the main critic ("not motivated", "importance unclear") to leverage its importance and demonstrate that it meets the standard of ICLR - and not the "red flag" (3 with high confidence of 5) that it should be rejected straight away :-)  We acknowledge that this was a major issue and fully assume our fault : this fully lies in the readability of the paper. In particular, due to our quite unconventional way of justifying our computational choices by the current knowledge of neurophysiological processes, we understand that it is not usual in the machine learning community. Still, we also think that --without changing the theoretical background of this work-- we can change its form to allow reviewers and the conference program chairs to converge to an optimal decision on the acceptance of this paper.

For that, we have highlighted the main contribution: homeostasis in one form or the other is necessary for dictionary learning. When enforcing a simple $\ell_2$ normalization, one may still obtain solutions for which some filters are more probable - and other more selective. Thanks to reviewers comments, we found a way to highlight this by reshuffling the atoms which we show: instead of selecting them randomly, we chose the extreme atoms (most and least probably selected). In hierarchical processing where the structural complexity of the features within each layer is preferentially homogeneous, it is undesirable to have a non uniform distribution of feature's structural complexity. We think that this knowledge from the biology of neural networks is an essential contribution to artificial networks.

Second, we have strengthened the qualitative evaluation of the algorithm. In the revision, we have shown results for larger datasets and longer learnings, highlighted the qualitative difference in the obtained dictionaries (just by changing the way they are displayed in the figure, see detailed responses). Importantly, we have included a supplementary material, which was absent from the first revision, and which shows the extension of this framework to other levels of sparsity, but also to different architectures. This was requested by the latest reviewer.

As a summary, we believe that these modifications were necessary to make the paper more impactful and we thank the reviewers for providing this essential input. We hope that this will allow us to present this quite unconventional work at ICLR.

---

### Submission · Authors · 2018-12-21
**Submission Withdrawn by the Authors**

I have read and agree with the withdrawal statement on behalf of myself and my co-authors.

---

### Meta-Review · Area_Chair1 · 2018-12-13
**Unclear what the benefit of this approach is**

**Confidence:** 5
**Recommendation:** Reject

**Metareview:**

This paper shows how to obtain more homogeneous activation of atoms in a dictionary. As reviewers point out, the paper is well written and indeed shows that the propose scheme results in a more uniform activation. However, the value of this contribution rests on making a case that uniformity is indeed a desirable outcome per se. As two reviewers explain, this crucial point is left unaddressed, which makes the paper too weak for ICLR.